# Landslide hazard cascades can trigger earthquakes

Zhen Zhang [1] ✉, Min Liu[1], Yen Joe Tan [1] ✉, Fabian Walter[2], Siming He[3], Małgorzata Chmiel [2,4] & Jinrong Su[5]

While earthquakes are well-known to trigger surface hazards and initiate hazard cascades, whether surface hazards can instead trigger earthquakes remains underexplored. In 2018, two landslides on the Tibetan plateau created landslide-dammed lakes which subsequently breached and caused catastrophic outburst floods. Here we build an earthquake catalog using machine-learning and cross-correlation-based methods which shows there was a statistically significant increase in earthquake activity (local magnitude ≤ 2.6) as the landslide-dammed lake approached peak water level which returned to the background level after dam breach. We further find that ~90% of the seismicity occurred where Coulomb stress increased due to the combined effect of direct loading and pore pressure diffusion. The close spatial and temporal correlation between the calculated Coulomb stress increase and earthquake activity suggests that the earthquakes were triggered by these landslide hazard cascades. Finally, our Coulomb stress modeling considering the properties of landslide-dammed lakes and reservoir-induced earthquakes globally suggests that earthquake triggering by landslide-dammed lakes and similar structures may be a ubiquitous phenomenon. Therefore, we propose that earthquake-surface hazard interaction can include bidirectional triggering which should be properly accounted for during geological hazard assessment and management in mountainous regions.

Mass movements such as landslides, avalanches, and debris flows are regarded as both major hazards and key forms of geomorphic evolution in mountainous regions[1–3]. These surface processes can be triggered by many factors including rainfall[4,5], snow melt[6,7], and human activities[8,9]. In seismically active regions, the risk is also heightened since large earthquakes can induce widespread mass wasting with devastating effects[10,11]. When these mass movements block rivers and form landslide-dammed lakes (LDLs), subsequent dam breaches may cause catastrophic outburst floods which significantly increase the degree and scope of the disaster[12,13]. For example, the 1933 M7.5 Diexi earthquake in the eastern Tibetan plateau directly caused ~7000 fatalities and triggered many large landslides, some of which dammed rivers. These LDLs breached in the following days, generating catastrophic outburst floods which then resulted in >2500 additional fatalities[14]. Therefore, the response of surface processes to earthquakes has been studied extensively, in particular the failure mechanisms of earthquake-triggered landslides[11,12], the quantification and prediction of the spatial distribution of mass movements[13,15], and the geomorphic evolution after earthquakes in mountainous areas[16,17], to improve our ability to mitigate the impact of these complex hazard cascades.

To date, studies about the interaction between earthquakes and mass movements have focused mainly on how earthquakes trigger

[1]Earth and Environmental Sciences Programme, Faculty of Science, The Chinese University of Hong Kong, Hong Kong S.A.R., China. [2]Swiss Federal Institute for Forest, Snow and Landscape Research, Zürich, Switzerland. [3]State Key Laboratory of Mountain Hazards and Engineering Safety, Institute of Mountain Hazards and Environment, Chinese Academy of Sciences, Chengdu, China. [4]Géoazur, OCA, Campus Azur du CNRS, Sophia Antipolis, Nice, France. [5]Earthquake Monitoring Center, Sichuan Earthquake Administration, Chengdu, China. ✉e-mail: zhenzhang@cuhk.edu.hk; yjtan@cuhk.edu.hk

mass movements and initiate hazard cascades, while a potential earthquake response to mass movements and their hazard cascades has received less attention. However, various studies have shown that processes which change the stress state of the Earth's crust by surface loading and/or fluid diffusion[18,19], such as earth tides[20], water storage behind dams[21], hydraulic fracturing[22], waste fluid disposal[23], and lake filling[24–27] can trigger/induce earthquakes when the stress increase on neighboring active faults exceeds a critical threshold[28,29]. Since hazardous mass movements and their complex hazard cascades are often accompanied by sediment redistributions and changes in water storage and associated surface loads[3,13], it is conceivable that the resulting stress changes can trigger earthquakes[30–33]. This would have important implications for hazard management in seismically active mountain regions.

Here, we use the 2018 Baige landslide hazard cascades (Fig. 1) on the Tibetan plateau as a case study to investigate earthquake triggering

by LDLs. We quantify the spatiotemporal evolution of seismicity and stress changes on a surrounding fault system due to this hazard cascade. We further use Coulomb stress modeling based on a global LDL database to explore the potential for LDLs to trigger earthquakes.

## Results
### 2018 Baige hazard cascades
At 22:05 China Standard Time (CST) on 10 October 2018, a massive landslide occurred at the Baige village on the Tibetan plateau[34–38] (Fig. 1 and S1) and deposited $2.5 \times 10^7$ m³ of sediments into the Jinsha River[35]. This created a landslide dam with ~1500 m length, >450 m width, and ~60 m height[35–37] (Fig. S1). With an upstream inflow of 1680 m³/s (ref. 35), the water level of the LDL gradually increased from ~2880 m asl to a peak of ~2932 m asl at 00:45 CST on 13 October (Fig. 2c), corresponding to inundation up to ~45 km upstream and a peak volume of ~$2.9 \times 10^8$ m³ (refs. 35–37). As the water naturally

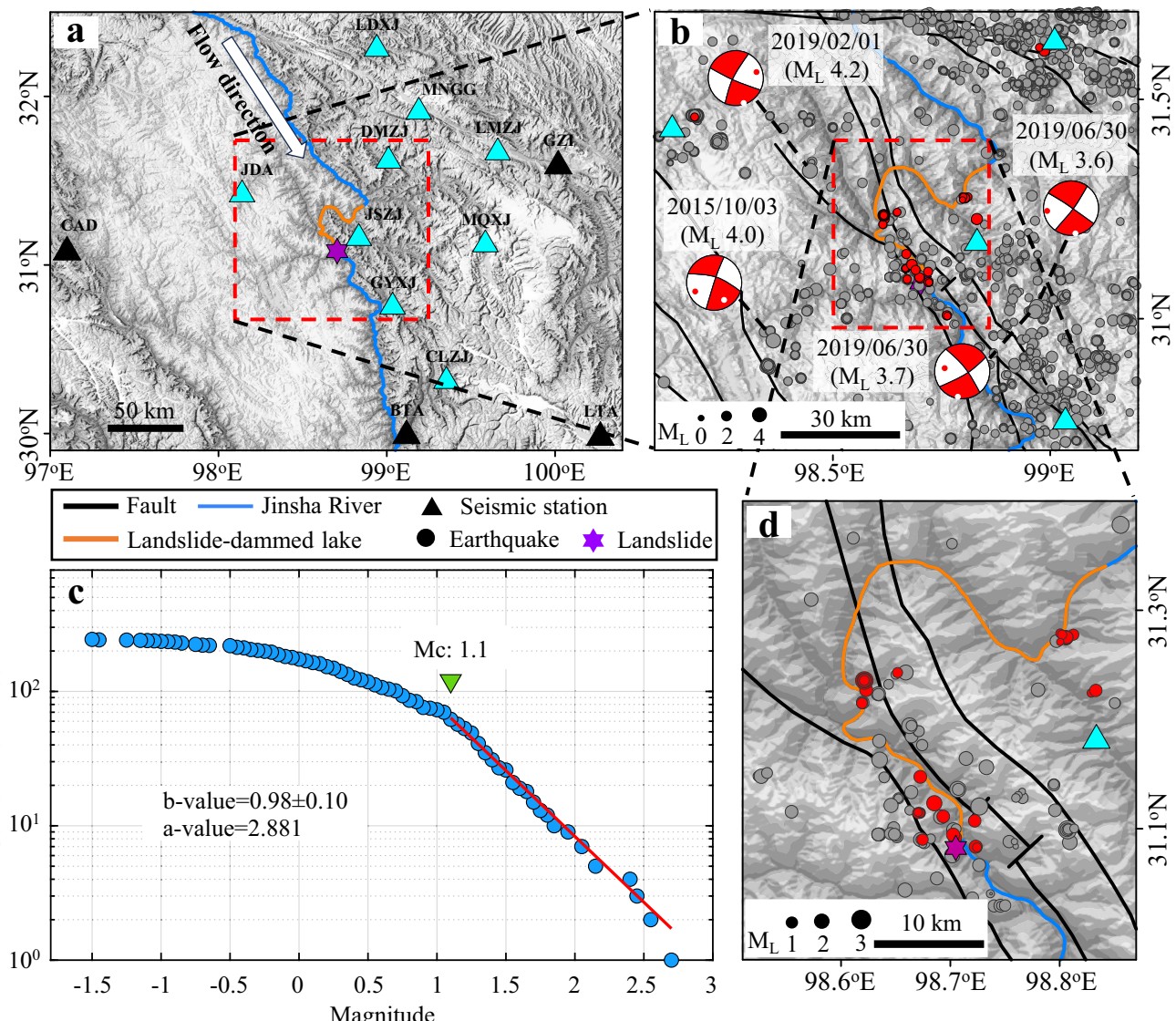

**Fig. 1 | Study site and regional seismicity. a** Distribution of seismic stations operating since January 2014 (black triangles) and May 2018 (cyan triangles). Red box marks the specific study region where we developed the earthquake catalog. **b** Seismicity map from January 2014 to January 2023 (gray filled circles) showing strike-slip focal mechanisms (red/white circles) available from the National Earthquake Data Center[39] plus earthquakes that occurred during the week (10 to 16 November 2018) when the second landslide-dammed lake (LDL) reached peak water level (red filled circles). Red box marks the region shown in (**d**). **c** Magnitude-frequency distribution of earthquakes from May 2018 to January 2023 within 10 km of the LDLs. Magnitude of completeness (Mc) and *a*- and *b*-values were calculated using the ZMAP software[40]. **d** Distribution of earthquakes from May 2018 to January 2023 (gray filled circles) within 10 km of the LDLs plus earthquakes that occurred during the week (10 to 16 November 2018) when the second LDL reached peak water level (red filled circles).

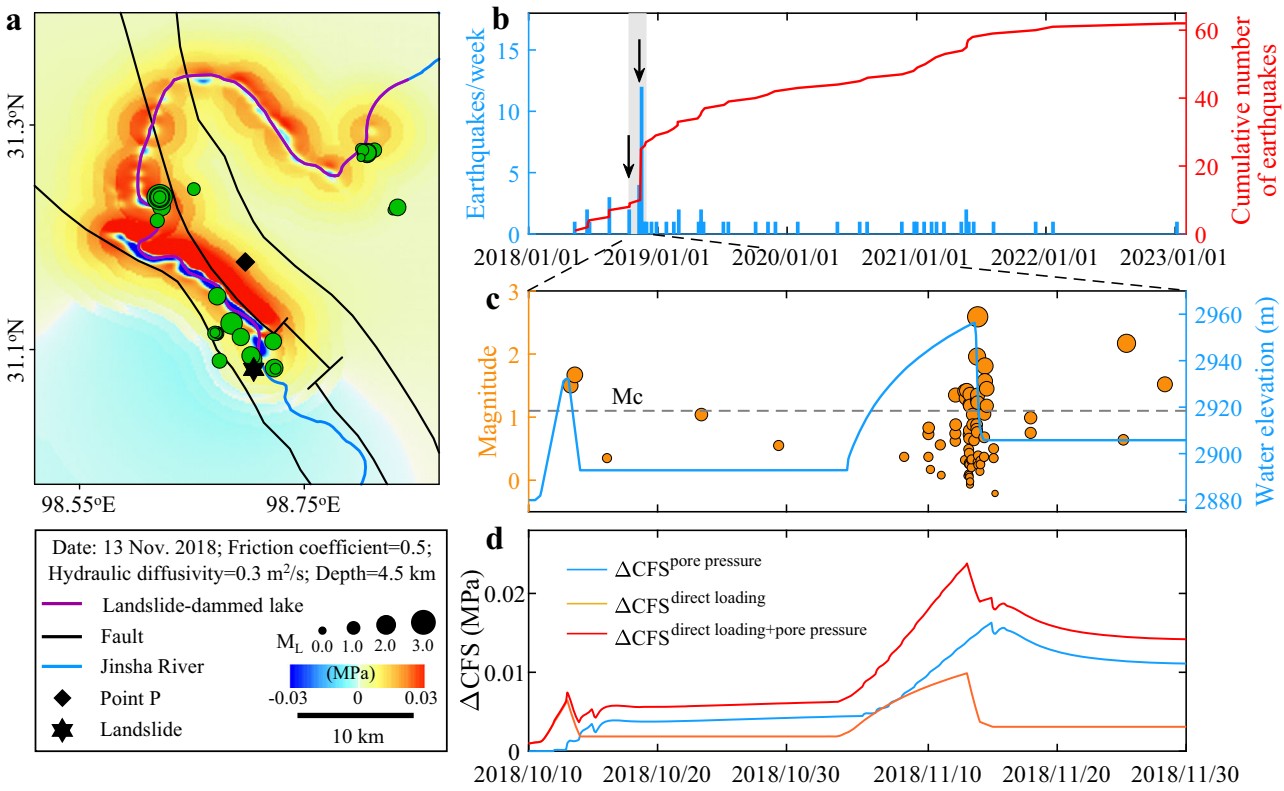

**Fig. 2 | Earthquakes triggered by the 2018 Baige landslide-dammed lake.**
**a** Spatial distribution of Coulomb stress change (ΔCFS) at depth of 4.5 km corresponding to −0.5 km asl on 13 November 2018 and earthquakes from 10 to 16 November 2018 (green filled circles) within 10 km of the landslide-dammed lakes (LDLs). ΔCFS is from the combined effect of direct gravitational loading and pore pressure diffusion (Fig. 2d). **b** Cumulative number of earthquakes with local magnitudes greater than or equal to the magnitude of completeness (Mc = 1.1) and weekly seismicity rate over a 5-year period with timings of the two 2018 Baige landslides (black arrows). Gray bar marks the period shown in (**c** and **d**). **c** Earthquake magnitudes and measured LDL water level[36,37] over a fifty-day period. Gray dashed line marks Mc. **d** Temporal evolution of ΔCFS at point P (**a**).

overflowed the west of the dam, the landslide dam breached rapidly. The outburst flood's discharge reached a peak of 10,000 m³/s at 06:00 CST on 13 October while the LDL's water level rapidly decreased. Eight hours later at 14:00 CST, the discharge was reduced to that of the upstream lake inflow. However, due to the residual dam, the water level upstream of the dam (~2893 m asl) remained higher than before the landslide.

About three weeks later at 17:40 CST on 3 November 2018, materials at the trailing edge of the first landslide's scar suddenly failed and the resulting landslide deposited an additional 8.7 × 10⁶ m³ of sediments on the residual dam formed by the first landslide[35–37]. This formed a new >96 m high landslide dam and a new LDL[35–37]. To mitigate the potential impact of another outburst flood, the Chinese government trenched an artificial spillway of 220 m length, 15 m depth, and >3 m width on the west side of the dam[36,37]. Water entered the spillway at 04:45 CST on 12 November marking the initiation of a second outburst flood and the LDL reached a peak water level of 2957 m asl at 13:40 CST on 13 November (Fig. 2c), corresponding to inundation up to ~70 km upstream and a peak volume of ~5.8 × 10⁸ m³. At 18:00 CST on 13 November, the outburst flood's discharge reached a maximum of 31,000 m³/s (refs. 35–37). Fourteen hours later at 08:00 CST on 14 November, the discharge decreased to the upstream inflow of ~600 m³/s. Like for the first LDL outburst, the water level upstream of the dam (2906 m asl) remained higher than before the landslide due to the residual dam (Fig. 2c).

## Seismicity rate increase after Baige LDLs' formation
Using continuous seismic data recorded by 13 nearby seismic stations from January 2014 to January 2023, we apply machine-learning and cross-correlation-based methods to build an earthquake catalog which contains ~3970 earthquakes with documented magnitude within a region of -1° × 1° (Fig. 1b). 244 of these earthquakes within 10 km of the LDLs occurred after May 2018 when a denser seismic network began operation. It is well-established that earthquakes can trigger surface hazards and initiate hazard cascades[5–7,12]. However, the Baige landslides were not triggered by earthquakes[34–38] as no earthquakes with magnitude $M_L$ ≥ 3.5 were recorded within 50 km in the 3 preceding years[39] (Figs. 1 and 2). Instead, in the week (10 to 16 November) when the second LDL approached its peak water level, 61 earthquakes with local magnitude up to ~2.6 occurred within 10 km of the LDLs.

As the second LDL approached peak water level, the number and magnitude of earthquakes started increasing as the water level rose and then peaked together with the water level peak (Fig. 2 and S2). Subsequently, as the LDL water level decreased following the dam breach, the earthquake activity gradually decreased back to the background rate. 45 of the 61 events in this earthquake sequence occurred before the largest magnitude ($M_L$ 2.6) earthquake, and other $M_L$ > 2.6 earthquakes during our observation period all had fewer than 4 aftershocks in the following week (Fig. S3). Therefore, this earthquake sequence is unlikely to be primarily an aftershock sequence of the largest magnitude event. We further verified that only 5 earthquakes (all with $M_L$ < 2.0) occurred within 60 km of the LDLs besides this earthquake sequence as the second LDL approached peak water level (Fig. 1b), and the largest earthquake that occurred within 60 km of the LDLs in the 6 months before this earthquake sequence has $M_L$ 2.7 (Fig. S4). Therefore, the earthquake sequence within 10 km of the LDLs is unlikely to have been triggered by surrounding large earthquakes.

We estimate the magnitude of completeness (Mc) to be 1.1 (ref. 40; Fig. 1c) for the earthquakes within 10 km of the LDLs. There are 62 earthquakes between May 2018 and January 2023 with local magnitude $M_L \geq 1.1$, 16 of which occurred in the week when the second LDL approached its peak water (Table S1). We then perform declustering using the Reasenberg method[41], leaving us with 54 events (Fig S5), 9 of which occurred in the week the second LDL approached its peak water level (Fig. S5, Table S1). In comparison, there were only 8 earthquakes in the previous 27 weeks (seismicity rate of ~0.3 events/week) and 37 earthquakes in the subsequent ~224 weeks (seismicity rate of ~0.2 events/week) in this region (Table S1). For each of ≥Mc and declustered catalogs, we consistently find that the increase in seismicity rate is statistically significant at a > 99% level based on both the statistic *P* (ref. 42) and statistic *Z* (ref. 43) tests (Methods). We further confirm that the increase in seismicity rate is also statistically significant at a >99% level (Table S1) based on both the improved statistic based on Poisson probability[44] and an empirically derived statistic[45] (Fig. S6) tests (Methods) which were developed to determine statistically significant changes in earthquake rate under small background rate. Hence, the significant increase in seismicity rate starting when the second LDL approached peak water level (Fig. 2 and S2) suggests that the two are related.

## Stress changes caused by Baige LDLs

We further investigate whether the LDLs could have triggered the earthquakes by modeling the Coulomb failure stress changes (ΔCFS) on the surrounding fault systems. In our study region, there are three local faults Boluo-Tongmai, Jinshajiang, and Gangtuo-Yidun faults

(from west to east) with similar surface traces of ~N50°W (ref. 46; Fig. 1a) though the exact fault dip and slip directions are unknown. Nevertheless, since January 2014, four M > 3.5 earthquakes occurred at distances between 16 and 35 km from the LDLs and have similar left-lateral strike-slip focal mechanisms with strikes that are also ~N50°W and dips of ~75°NE (ref. 39; Fig. 1a). In addition, we find that the first motions recorded by surrounding seismic station of the two largest earthquakes ($M_L$ 2.6 and 1.9) as the second LDL approached peak water level are generally consistent with the left-lateral strike-slip focal mechanisms of these surrounding large earthquakes (Fig. S7). Therefore, we take these as our receiver fault geometry for the ΔCFS modeling (Methods). Furthermore, the receiver faults are assumed to be planar and located at 4.5 km depth which is deeper than 95% of our observed seismicity (Fig. S8) and corresponds to ~−0.5 km asl because ~80% of our study area is at elevation between 4 and 5 km asl[35].

We find that as the LDLs' water level increased following the landslides, the ΔCFS in the seismic region generally increased due to both the direct gravitational loading of the LDL and pore pressure diffusion, while ΔCFS decreased when the LDLs' water level decreased following the dam breaches (Fig. 2 and S9). The larger and longer-lasting LDL generated greater ΔCFS. The ΔCFS due to the first, smaller, and shorter-duration LDL peaked at 0.007 MPa which is smaller than the stress increase of >0.01 MPa typically associated with stress triggering of earthquakes (refs. 47,48; Fig. 2d). This could explain why there was no significant increase in seismicity rate associated with the first LDL's formation (Fig. 2c). In comparison, about 3 weeks later on 3 November, the second landslide formed a larger LDL. The ΔCFS due to this second LDL peaked at 0.024 MPa when the LDL's water level

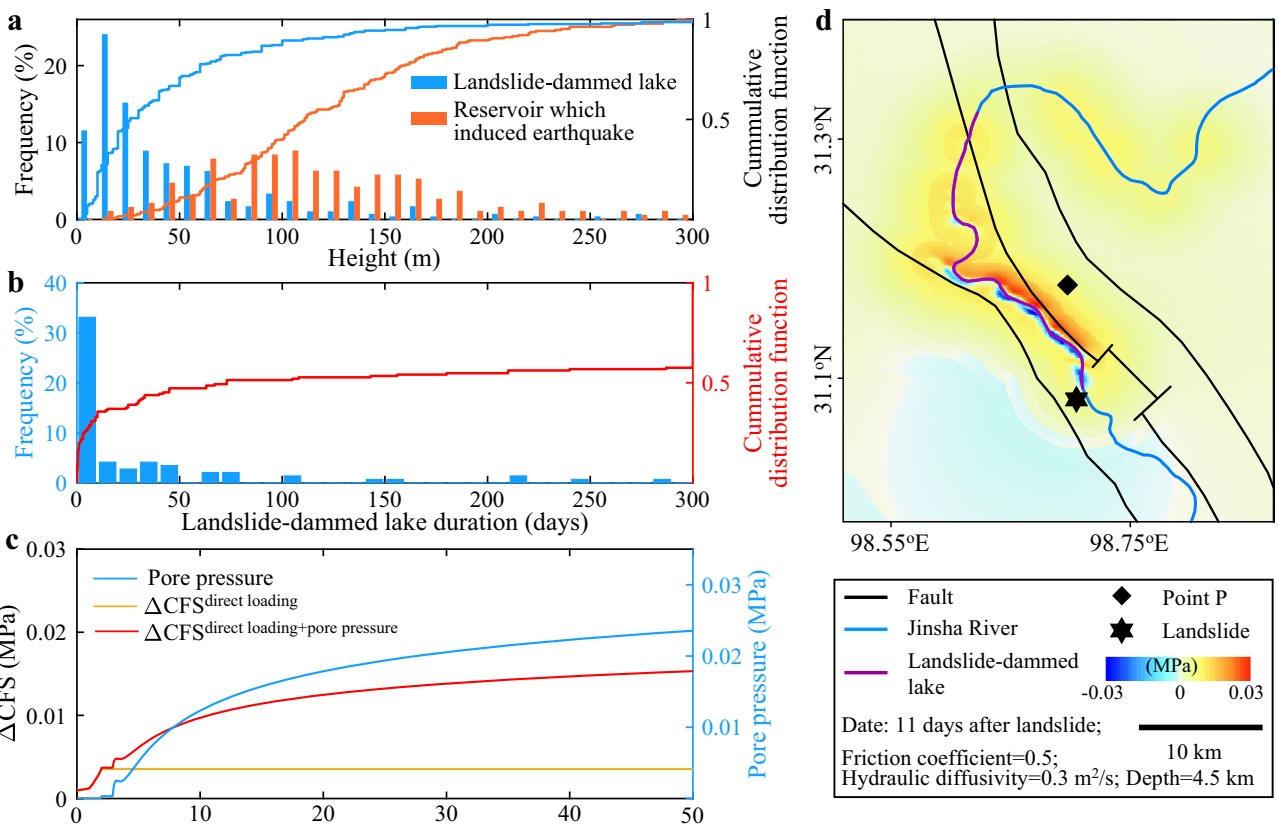

**Fig. 3 | General properties of landslide-dammed lakes (LDLs). a** Frequency distribution and cumulative distribution functions of the height of reservoirs documented to have induced earthquakes[49] and landslide-dammed lakes[50] worldwide. **b** Frequency distribution and cumulative distribution function of the duration of landslide-dammed lakes worldwide[50]. **c** Temporal evolution of ΔCFS and pore pressure at point P (**d**) for a 30-m deep LDL. **d** Spatial distribution of ΔCFS at depth of 4.5 km corresponding to −0.5 km asl, 11 days after the formation of a 30-m deep LDL. ΔCFS is from the combined effect of direct gravitational loading and pore pressure diffusion.

approached its peak, coinciding with maximum earthquake activity (Fig. 2c). Although the subsequent dam breach lowered the LDL's water level which decreased the ΔCFS due to direct gravitational loading, the pore pressure diffusion effect kept the ΔCFS elevated which might explain why earthquake activity continued over the next few days (Fig. 2c).

While direct loading resulted in a decrease in ΔCFS directly below the LDL, it resulted in a slight increase in ΔCFS in surrounding regions (Fig. S9a). On the other hand, fluid diffusion significantly increased the pore pressure around the LDL, and the ΔCFS in regions closer to the LDL were larger (Fig. S9b). Therefore, due to the combined effect of direct loading and pore pressure diffusion, the ΔCFS around the LDL increased, especially on its east side. We find that ~90% of the earthquakes locate within regions of positive ΔCFS (Fig. 2a). The close spatial and temporal correlation between the calculated positive ΔCFS and the statistically significant increase in earthquake activity suggest that the earthquakes were likely triggered by the LDL.

### Earthquake triggering by LDLs

Considering the fundamental mechanism of earthquake triggering by LDLs and reservoir-induced earthquakes is similar, we further explore the potential for other LDLs to trigger earthquakes. Based on HiQuake, currently the most complete and up-to-date freely available database of ~226 cases of reservoir-induced earthquakes spanning the period 1933-2019 globally[49], we find that the minimum and median heights of ~190 reservoirs with documented height that induced earthquakes are ~13 and 110 m, respectively (Fig. 3a). In comparison, from a global database of ~410 LDLs with dams >1 million m³ in volume reported worldwide spanning the period 1900–2018 (ref. 50), we find that of the ~300 LDLs with documented dam height, ~73% and ~10% were higher than the minimum and median heights of reservoirs which induced earthquakes, respectively (Fig. 3a). Therefore, some LDLs globally are of similar heights as reservoirs which induced earthquakes.

We further calculate the stress response of surrounding faults in our study area to an LDL with a depth of 30 m which is the median dam height in the global LDL database[50] (Fig. 3b). In this case, the ΔCFS at a given point (Figs. 2a and 3d) would exceed 0.01 MPa after ~11 days due to direct loading and pore pressure diffusion if the dam does not breach (Fig. 3c). The stress responses to this typical LDL at different sites can vary (Fig. 3d) and depend on factors such as the fault types[51] (Fig. S10), geometries (Fig. S11), and depths (Fig. S11). Nevertheless, our modeling shows that both the fault types and geometries primarily affect stress distribution rather than amplitude of ΔCFS (Figs. S10 and S11), and stress changes caused by the LDLs are larger at shallower depths (Fig. S11). Considering ~64% of the ~145 LDLs in the global database with documented duration lasted more than 11 days before dam breach (Fig. 3b) with no correlation with dam height (Fig. S12), our modeling suggests that other LDLs globally have the potential to trigger earthquakes if there are critically stressed faults nearby.

## Discussion

While gravitational loading can cause Coulomb stress to decrease in some areas, pore pressure always causes an increase in ΔCFS with its amplitude gradually decreasing away from the LDLs (Fig. S9). For earthquakes that occurred in regions where gravitational loading resulted in a decrease in Coulomb stress, it is clear that pore pressure is the primary mechanism triggering these events, though we neglect the coupling effect between pore pressure diffusion and gravitational loading[24–27]. However, for those earthquakes that occurred in regions where gravitational loading resulted in an increase in Coulomb stresses, both gravitational loading and pore pressure may jointly control the triggered seismicity. After the second LDL formed, while the water level rises and seismicity begins to occur from 10 to 13 November (Fig. S2), the relative contribution to ΔCFS of gravitational loading decreases gradually from 0.45 to 0.41 (Fig. S13). Subsequently, after the dam breached on 13 November which is the day of peak seismicity rate, the water level rapidly decreased and the relative contribution to ΔCFS of gravitational loading decreases sharply from 0.41 to 0.19 so pore pressure becomes the dominant triggering mechanism. However, the relative contribution to ΔCFS of pore pressure depends on the assumed hydraulic diffusivity (Fig. S14). In addition, since the pore pressure diffusion is from a long winding river (LDLs) instead of a point source, the seismicity do not display obvious migration pattern with time (Fig. S15).

The region surrounding the 2018 Baige LDLs is seismically not very active which allowed us to identify the sudden, sharp increase in earthquake activity and link it to the LDLs. However, this might explain why the number and magnitude of triggered earthquakes are relatively

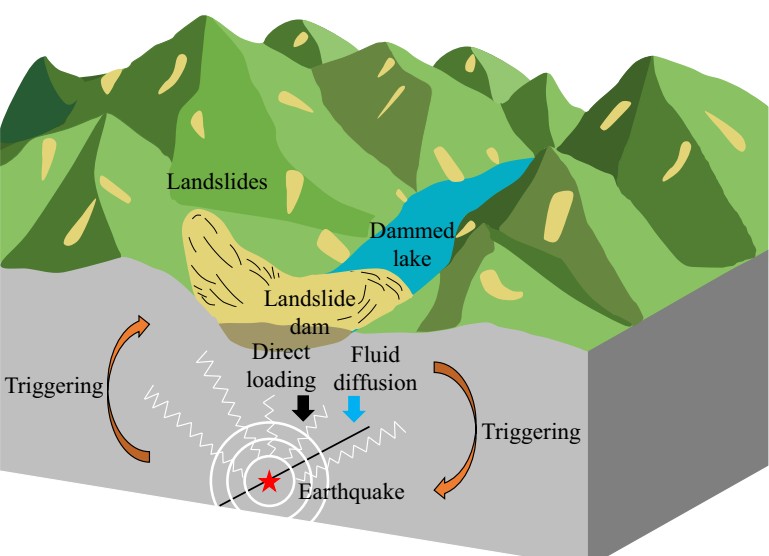

**Fig. 4 | Schematic representation of earthquake-surface hazards interaction.** Previous studies[5,6,13,16] had shown that earthquakes can trigger landslides which can block rivers and form landslide-dammed lakes. We show that direct gravitational loading and pore pressure diffusion from landslide-dammed lakes can in turn increase stresses on surrounding faults and trigger earthquakes. The interaction between these hazards is thus bidirectional.

small since earthquake triggering also depends on other factors such as the availability of critically stressed faults. Since cases of reservoir-induced earthquakes are relatively common and have included M > 7 earthquakes[49], LDLs in seismically more active regions can potentially trigger more and larger earthquakes, assuming they increase ΔCFS given the fault locations and orientations. For example, the 2008 M7.9 Wenchuan earthquake triggered more than 100,000 landslides and formed a few hundred LDLs[52–54]. Some landslides were >30 times larger than our landslides[53] and the largest LDL lasted for ~1 month and reached a maximum depth of ~82 m before breaching[54]. Therefore, we suggest that such LDLs have greater potential to trigger earthquakes and the earthquake-surface interaction could even form a feedback loop (Fig. 4). Furthermore, while reservoir planning aims to avoid seismically active regions, ~20% of LDLs are estimated to have formed by earthquake-triggered mass wasting[50]. Consequently, earthquake triggering by LDLs might be a ubiquitous phenomenon in seismically active regions though at least to some extent, this effect may be masked by earthquake-earthquake triggering in such regions[55].

While we only demonstrated earthquake triggering by LDLs, other surface hazards may similarly alter the stress state of surrounding faults and trigger earthquakes. For example, glacial lakes whose volumes are increasing due to climate change[56] can alter the stress state of surrounding faults through both direct gravitational loading and pore pressure diffusion. Therefore, our results suggest that such bidirectional interaction between surface hazards and earthquakes should be properly accounted for in future risk assessment of geological hazards and hazard management in mountainous regions.

## Methods

### Earthquake catalog building

We adopt an AI-based workflow to develop a new earthquake catalog from 2014 to 2023 based on the 13 permanent seismic stations[57] (Fig. 1). We first use the machine-learning-based detector, PhaseNet[58], to identify P- and S-wave arrival times from the continuous waveforms. Subsequently, these P- and S-wave arrival times are associated into individual earthquakes using a high-throughput seismic phase associator, PyOcto[59], and a local 1-D velocity model[60]. A threshold of at least two P picks, one S pick, and a total of five P and S picks is adopted during the PyOcto association, resulting in 6692 earthquakes within a region of 2.5° × 3° (latitude: 30° to 32.5°; longitude: 97° to 100°). We then manually inspect the waveforms of 859 earthquakes around our target area (latitude: 30.7° to 31.7°; longitude: 98.1° to 99.2°) to exclude 112 non-tectonic or unreasonably associated events. We then use the absolute location method Hypoinverse[61] to further refine the hypocenters of the remaining 747 events and calculate their location uncertainties.

We also estimate the local magnitudes for all retained earthquakes based on their S-wave amplitudes and a recently improved national standard magnitude scale that is specific to the Sichuan region, China[62,63]. The maximum amplitudes of horizontal component waveforms are measured after deconvolving the instrument response from the raw waveforms and then convolving the obtained signal with the theoretical Wood–Anderson seismometer response. The measured waveform window starts 0.5 s before the P wave arrival and is twice the predicted S–P travel time in length. Note that we only estimate magnitudes for 669 events with at least two stations having signal-noise ratios of S waves larger than 3. The estimated magnitudes range from $M_L$ 0.1 to 4.2.

To further improve the magnitude of completeness, we use a template matching technique, Graphics Processing Unit-based match and locate technique (GPU-M&L)[64], to detect and locate more earthquakes based on the 669 retained earthquakes with estimated magnitude. P- and S-waves of these events are simultaneously used in GPU-M&L. The length of template window is 6 s starting 1 s before the P- and S-wave arrivals picked by PhaseNet. We filter the continuous and template waveforms from 2 to 8 Hz. The 1-D velocity model used in the above absolute earthquake location was adopted in GPU-M&L as well. Note that in GPU-M&L, we relocate the epicenters of detections through a grid search method with searching space 1.2 × 1.2 km in the horizontal direction and searching interval of 0.06 km if template and continuous waveforms were recorded by at least three seismic stations. Otherwise, the locations of detections will be defined as that of the templates. By manually inspecting these detections, the detection threshold is defined as correlation coefficient of 0.3 and 12 times median absolute deviation (MAD) if the common component number between template and continuous waveforms is ≥ 9. Otherwise, the detection threshold is defined as correlation coefficient of 0.5 and 15 times MAD. In total, GPU-M&L resulted in 4294 detections including the 669 template events. The magnitudes of the newly detected earthquakes are estimated based on the local magnitudes of the templates and the amplitude ratio between these detections and their corresponding templates[65]. Note that due to the poor local seismic station coverage with available data before May 2018, 22 earthquakes that occurred from March 2014 to May 2018 in the routine catalog, which utilized data from the entire China, were not detected. Thus, these missing earthquakes are manually added to our earthquake catalog.

### Declustering and seismicity rate change analysis

Using the approach from Reasenberg[41], we identified 5 clusters of earthquakes containing a total of 13 events (5 main earthquakes and 8 foreshocks/aftershocks) out of 62 events which occurred over ~5 years, using the following input parameters[41]: look-ahead time from 0.5 to 3 days, confidence probability of 0.9, effective min magnitude cutoff of 1.1, the increase in lower cutoff magnitude during clusters of 0, and the number of crack radii surrounding each earthquake of 5.

We then evaluate the statistical significance of the observed seismicity rate change based on statistic $P$ (ref. [42]) and statistic $Z$ (ref. [43]). The statistic $Z$ is defined as follows:

$$Z = \frac{N\triangle t_M - M\triangle t_N}{\sqrt{N\triangle t_M^2 + M\triangle t_N^2}} \qquad (1)$$

Where $N$ and $M$ represent the number of earthquakes after and before an event in time windows $\triangle t_N$ and $\triangle t_M$, respectively. When $N$ and $M$ are sufficiently large and in the null hypothesis of no seismicity rate change, $Z$ follows a Gaussian distribution with zero mean and unit standard deviation. Statistic $Z$ was first proposed by Habermann[43] to evaluate the statistical significance of seismicity rate changes. After declustering, there were 9 earthquakes in the week (10 to 16 November) when the second LDL approached its peak water level. There were only 8 earthquakes in the previous 6 months (~27 weeks) which translates to a seismicity rate of 0.3 events/week. Based on Eq. 1, we obtain a $Z$ value of 2.90 taking $N$, $M$, $\triangle t_N$, and $\triangle t_M$ as 9, 8, 1, and 27, respectively. A change is considered statistically significant if $|Z| > 2$ (ref. [43]), and our observed seismicity rate increase after declustering[41] is statistically significant at a 99.62% level based on statistic $Z$.

In the null hypothesis that the seismicity rate in the week after (Poisson process with mean rate $\lambda_N$) and the week before (Poisson process with mean rate $\lambda_M$) an event is the same, the statistic $P$ reads:

$$P\left(\frac{\lambda_N}{\lambda_M} > 1\right) = 1 - \frac{1}{N!M!}\int_0^\infty e^{-x}x^M \Gamma(N+1,x)dx \qquad (2)$$

Where $\Gamma(n,x) = \int_0^x e^{-t}t^{n-1}dt$ is the incomplete Gamma function and follows a uniform distribution between 0 and 1 (ref. [42]). 1-$P$ gives us the probability that an observed value could be obtained by chance if the null hypothesis of no rate change is true[42]. While there were no earthquakes in the ~1 month before the second LDL, we still take the

seismicity rate for the week before the second LDL approached its peak water level as 0.3 events/week based on the average seismicity rate in the previous 6 months to be conservative. Based on Eq. 2, we obtain a $P$ value of 0.9982 taking $N$ and $M$ as 9 and 0.3, respectively. This implies that the probability that our observed seismicity rate increase occurred by random chance is 0.18%. Therefore, our observed seismicity rate increase after declustering[41] is statistically significant at a >99.82% level based on statistic $P$.

We further evaluate the statistical significance of the observed seismicity rate change based on both an improved statistic based on Poisson probability[44] and an empirically derived statistic[45] which were developed to determine statistically significant changes in earthquake rate under small background rate. We assumed that earthquakes occur independently at a constant rate, following a Poissonian distribution[44]. To determine the statistical significance of seismicity rate increase, we compare the number of expected earthquakes in a 1-week window to the number of earthquakes in the 1-week window after the trigger and calculate the Poisson probability of obtaining the number of earthquakes ($\nu$) in the 1-week window after the trigger given the expected number of events ($\mu$) in the 1-week window.

$$P_\mu(\nu) = e^{-\mu} \frac{\mu^\nu}{\nu!} \qquad (3)$$

For this case of low background seismicity rate, similar to the idea of ref. 44, we conservatively regard the largest number of earthquakes (3) in a 1-week window over our observation period, except for this earthquake sequence, as the expected number of events ($\mu$) in a 1-week window. Based on Eq. 3, we obtain a $P$ value of 0.0027 taking $\mu$ and $\nu$ as 3 and 9, respectively. This implies that our observed seismicity rate increase after declustering[41] is statistically significant at a >99.73% level.

We also use the empirical statistical method of ref. 45 to determine the significance in the seismicity rate change. First, we count the number of earthquakes in each 1-week time window incorporating a 1-day sliding window. There are >1700 windows from May 2018 to January 2023. We then assign the number of earthquakes $N_{count}$ within each window to a timestamp at the start of the window. We next build a histogram of $N_{count}$ values (Fig. S6). Finally, using percentile measurements, the >99% statistically significant level is 3 earthquakes in a 1-week window. We observed 9 earthquakes (after declustering) as the second LDL approached the peak water level, hence this seismicity rate increase is statistically significant at a >99% level.

We also tested the impact of different input parameters on the declustering based on the approach from Reasenberg[41]. We identified 4 clusters of earthquakes containing a total of 16 events (4 main earthquakes and 12 foreshocks/aftershocks) out of 62 events which occurred over ~5 years, using other suggested input parameters[41]: look-ahead time from 1 to 10 days, confidence probability of 0.95, effective min magnitude cutoff of 1.5, the increase in lower cutoff magnitude during clusters of 0.5, and the number of crack radii surrounding each earthquake of 10. While the declustered earthquake catalogs with different input parameters are different, we further confirm that this seismicity rate increase is statistically significant at a > ~90% level based on the four above statistical tests (Table S1).

## Coulomb stress change calculation

Coulomb stress change ($\Delta$CFS) on the surrounding fault systems due to the LDL is a result of the combined effect of direct gravitational loading and pore pressure diffusion. We model $\Delta$CFS due to both direct loading and pore pressure diffusion using the GeoTaos software[66]. Since the direct gravitational loading effect can be regarded as the result of point forces acting vertically on the surface of a homogenous elastic half-space, the loading from the LDL can be calculated by the convolution of the Green's function and the distributed surface forces. The distributed surface forces can be estimated based on the spatiotemporal evolution of the LDL's water level. To quantify the spatiotemporal evolution of the LDL's water level, the LDL is mapped as a series of square cells with a size of 200 by 200 m. The elevation of each square cell is estimated from the GDEM V3 30 m. Based on the elevation of each cell, the water depth of each cell can then be estimated. Therefore, the distributed surface forces and thus the direct gravitational loading effect of each cell can be calculated using the water depth of each cell. Note that we ignore the $\Delta$CFS resulting from gravitational loading of landslide sediments, since these landslide sediments are not a new source of mass but only the very local redistributions (~1–2 km) of sediments along the landslide sliding path.

## Uncertainty analyses

Before more seismic stations became operational in May 2018, no earthquakes were detected within 10 km of the LDLs due to limited station coverage and/or low seismicity rate. Nevertheless, in the ~250 weeks of observation period since May 2018 for which we have an improved earthquake catalog, 61 earthquakes occurred within 10 km of the LDLs during the week when the second LDL's water level almost peaked while there were only 183 earthquakes over the ~247 weeks outside the time period of the two LDLs i.e., a background seismicity rate of only ~0.7/week over ~5 years (Table S1). Therefore, our catalog shows that the seismicity rate during the LDL is significantly higher than the background seismicity rate of the region. We chose to focus on earthquakes that occurred within 10 km of LDLs since the region with stress increase of >0.01 MPa is always located within 10 km of the LDLs (Fig. 2a). Nevertheless, the earthquake activity within larger and smaller regions shows a similar accelerated trend during the landslide hazard cascades (Fig. S16).

As the second LDL approached peak water level, ~50% and ~95% of the hypocenters in the earthquake sequence were at depths of <1.2 and <4.5 km, respectively (Fig. S8). Similarly, for the earthquake catalog before template matching, the median depth of these triggered earthquakes is ~1.2 km (Fig. S8). However, the mean horizontal and vertical location uncertainties of our cataloged earthquakes are ~4 and ~8 km, respectively. To further constrain the likely depths of these events, we look at the larger earthquakes which are more accurately located since they are recorded by more stations. We find that the median depth of all $M_L > 1.1$ earthquakes during the second LDL is ~1.5 km with average depth uncertainty of ~6 km. Our modeling shows that $\Delta$CFS down to a depth of 7 km can reach levels that trigger earthquakes (Fig. S11d), and the $\Delta$CFS is larger at shallower depths (Fig. S11). Furthermore, ~60% of reservoirs with documented earthquake depths have induced earthquakes up to ~8 km depth (Fig. S17). Therefore, we conclude that the earthquake sequence can be triggered by stress changes due to the LDL.

The stress change on surrounding faults due to the LDLs partly depends on the fault geometry. In contrast to the well-documented fault strike, the dip of the fault is less well constrained. We assume a fault dip of 75°NE based on the focal mechanisms of four M > 3.5 earthquakes nearby and the consistency of the first motions of the two largest earthquakes (ML 2.6 and 1.9) as the second LDL approached peak water level with these focal mechanisms (Fig. S7), though we cannot confirm that the focal mechanisms of the smaller earthquakes necessarily share similar geometries. Nevertheless, our modeling shows that the fault dip mainly affects the spatial distribution and amplitude of $\Delta$CFS but not its sign, and the seismicity generally locates within regions of positive $\Delta$CFS for a range of different assumed fault dips (Fig. S11e–h). Hence, the dip uncertainty has minimal effect on our conclusion that landslide hazard cascades can trigger earthquakes.

The stress change on surrounding faults caused by LDL is also related to the fault friction coefficient and the fluid diffusion

**Article**

coefficient. We assumed a friction coefficient of 0.5 for our ΔCFS modeling. However, the chosen friction coefficient has minimal effect on both the spatial distribution and amplitude of ΔCFS (Fig. S18). In addition, due to differences in rock properties and depth-dependent confining pressure, crustal hydraulic diffusivity has been found to range from 0.01 to 5 $m^2$/s (refs. 67,68). Both the geometry and closure of faults and fractures also cause significant spatial variation in hydraulic diffusivity[69,70]. These factors make it challenging to obtain a precisely constrained hydraulic diffusivity for the calculation of ΔCFS. Nevertheless, higher fluid diffusivity causes the pore water pressure to increase faster, which increases the amplitude of ΔCFS over a wider area. Our chosen value of 0.3 $m^2$/s is on the lower end of observed values and is a typical value observed in fluid-driven seismic swarms in different regions[71–73]. While hydraulic diffusivity strongly influences the time at which ΔCFS exceeds a certain threshold, we further confirm that ΔCFS exceeds 0.01 MPa for hydraulic diffusivities ranging from 0.02 to 2 $m^2$/s. On the day of peak seismicity rate, the relative contribution to ΔCFS of pore pressure increases from -0.1 to 0.9 as the assumed hydraulic diffusivity increases from 0.02 to 2 $m^2$/s (Fig. S14). We conclude that our main conclusion that LDLs can trigger earthquakes is robust to changes in the discussed parameter choices.

### Reporting summary

Further information on research design is available in the Nature Portfolio Reporting Summary linked to this article.

## Data availability

The reservoir data which induced earthquakes are available at the Human-Induced Earthquake Database (HiQuake) http://inducedearthquakes.org/. The database for landslide-dammed lakes is available from https://doi.org/10.1016/j.earscirev.2020.103116. The earthquake catalog produced in this study is available at https://doi.org/10.5281/zenodo.10807902. The GDEM data are provided by ASTER https://asterweb.jpl.nasa.gov.

## Code availability

The software PhaseNet[58] and PyOcto[59] used in this study are publicly available at https://doi.org/10.1093/gji/ggy423 and https://doi.org/10.26443/seismica.v3i1.1130, respectively. The software Hypoinverse[61] and GPU-M&L[64] are publicly available at https://www.usgs.gov/software/hypoinverse-earthquake-location and https://doi.org/10.1785/0220190241, respectively. The software Zmap[40] and GeoTaos[66] are publicly available at http://www.seismo.ethz.ch/en/research-and-teaching/products-software/software/ZMAP and http://bemlar.ism.ac.jp/lxl/Taos/Download.htm, respectively.

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

## Acknowledgements

We thank Jinping Zi, Peifeng Wang, and Yiyuan Zhong for valuable discussions. This work was supported by the Croucher Tak Wah Mak Innovation Award (Y.J.T.), CUHK Direct Grant for Research (Grant 4053594; Y.J.T.), CUHK Research Fellowship Scheme (Grant 4200720; Y.J.T.), CUHK Postdoctoral Fellowship Scheme (Grant 3135043; Y.J.T.), the National Key Research and Development program of China (Grant No.2022YFF0800604; S.H.), and the Major Program of the National Natural Science Foundation of China (Grant No.42090051; S.H.).

## Author contributions

Conceptualization: Z.Z. and Y.J.T.; Investigation: Z.Z., M.L., Y.J.T., F.W., S.H., M.C., and J.S.; Methodology: Z.Z., M.L., and Y.J.T.; Project administration: Y.J.T. and S.H.; Supervision: Y.J.T. and S.H.; Writing-original draft: Z.Z. and Y.J.T.; Writing-review and editing: Z.Z., M.L., Y.J.T., F.W., S.H., M.C., and J.S.

## Competing interests

The authors declare no competing interests.
