## [Peer Review File · Nature Communications]

Landslide hazard cascades can trigger earthquakesReviewers' Comments:

Reviewer #1 (Remarks to the Author):

- What are the noteworthy results?

This is generally a well written paper with a provocative suggestion, that landslides could potentially create a large enough of stress disturbance that will trigger earthquakes. This assertion is not far-fetched, as we see many are documenting induced seismicity throughout the world, from gas extractions, enhanced oil recovery, hydraulic fracturing, reservoir induced seismicity, and even seismicity from glacial retreat. However, this work depends heavily on the earthquake catalog that the authors did not produce, and I worry that all of uncertainties are not explored, making the case very weak. In its current state, I cannot recommend this paper for publication.

- Does the work support the conclusions and claims, or is additional evidence needed?

The previously existing seismic network was enhanced in May 2018, and because of this, the sensitivity of the network changed, allowing for more small magnitude earthquakes to be detected and located. Thus, it is difficult to determine the background seismicity rate prior to the landslide in Nov. 2018, and the seismicity rate of the region in general. Furthermore, the authors do not plot the station distribution (before or after May 2018), and the depth uncertainties are not stated. If these events are deep in the crust, making the case for induced seismicity becomes very difficult. Even if they were shallow, we have no idea of the station distribution and true depth uncertainly, plus the diffusivity should be known to make any connection to water migration and the timing of the proposed induced seismicity. To have good depth determination, a seismic station must be within a focal depth to have any chance of constraining depth, plus having a well calibrated velocity model is critical for accurate depth estimates. There is no mention of the station distribution, the local velocity model, the location errors, and the number of stations that recorded the events. Furthermore, these are catalog events, and perhaps the seismicity is clustered near the landslide itself. Using an approach like HypoDD (double difference) might show that the events are clustered spatially, invalidating the argument that the events are spread throughout the region of inundation. None of these location approaches are explored; thus, the correlation with the landslide is tenuous at best. Lastly, all of the triggered events seem to be less than M1.4. These are micro earthquakes, and they are difficult to locate and analyze. The seismology for these earthquakes will have to be more clearly documented for this work to be convincing.

- Are there any flaws in the data analysis, interpretation and conclusions? Do these prohibit publication or require revision?

Although not under the author's control, the number and magnitude of earthquakes are small, and running statistics on these small numbers is difficult. The authors decluster the catalog and use a couple of statistical methods to determine a change in seismicity rate. Complicating this, associating the events to the water inundation caused by landslide proves difficult given the time delay (two weeks) before the

onset of seismicity. I would explore the use of other statistics used for dynamic triggering studies (works of Z. Peng, K. Pankow, R. Alfaro-Diaz, D. Kilb, E. Brodsky). If these events are indeed triggered, they would be classified as delayed triggering, which even is harder to show statistically. Finally, seismicity also appears widespread in the region, not necessarily on the faults. Again, this may be the result of poor locations. There is simply no way to assess given the information given in the study.

- Is the methodology sound? Does the work meet the expected standards in your field?

Stress modeling also depends on assumed faults. The fault information also appears not well documented (orientation of the faults are not known). Determining focal mechanism would help, but these are small earthquakes. Perhaps a composite focal mechanism can be used to determine what the stress direction is that would be released with increase pore pressure. Furthermore, how deep are these faults, and how long would it take for the water to reach these faults?

Reviewer #2 (Remarks to the Author):

Zhang et al. focuses on the potential impact of landslide-dammed lakes on the triggering of earthquakes on nearby faults. The paper is easy to read and addresses an interesting issue, which I feel deserves to receive the attention of a high-impact journal. Too few studies have indeed focused on the impact of surface processes during extreme events on the triggering of earthquakes and deformation. I was therefore thrilled to receive this paper for a review.

Unfortunately, I have found too many major weaknesses that at this stage prevent, in my opinion, the paper from being published. Despite that, I also feel that most of these major issues can be thoroughly addressed by the authors, and I therefore would ask the editor and authors to consider a phase of major revisions, but not necessarily of rejection (or otherwise with the opportunity for resubmission). I want to thank the authors for this interesting manuscript, and I hope they will receive most of these comments as a means to increase the quality of their manuscript, and not only as a criticism.

I first summarize below my main comments and then describe my other comments line-by-line.

With my best regards,

Philippe Steer

Main comments:

- False claim of the manuscript: The manuscript repeats several times (lines 14-15, 49-52, in conclusion as well) that this is the first time that a link between surface hazards/processes (or erosion or landslides) and the triggering of earthquakes is made. This is simply wrong and surprising to see this affirmation, as several visible papers in high-impact journals were published on this subject during the last 15 years:

Calais et al. (2010 - Nature), Steer et al. (2014 - Nature Communications), Steer et al. (2020 - Scientific reports), Jeandet Ribes et al. (2020 - GRL), and probably others. Moreover, several papers have already shown that lake water variation can impact seismicity and trigger earthquakes (Gough & Gough, 1970; Kebeasy & Gharib, 1991; Simpson et al., 2018; and many more). This paper is therefore not the first one to address the issue of the triggering of earthquakes because of surface hazards, landslides, or erosion, nor it is the first one to show the impact of lake water level fluctuation on earthquake triggering. The manuscript by Zhang et al. therefore needs to account for these previous studies, in particular in the title, introduction, discussion and conclusion, and to correct the importance of their study respectively to what was already demonstrated by these previous studies. This manuscript is however the first one (if I have not missed a previous article) to address the issue of the response of seismic activity to landslide-dammed lakes filling. But this is largely different from what the current manuscript claims.

- Issue with declustering: The manuscript uses two classical approaches to decluster the earthquake catalog they investigate. However, only 3 over 39 earthquakes are identified as aftershocks and removed from the catalog. This is a very unusually low number of earthquake removal by declustering. I identify two issues that need to be addressed by the manuscript: 1) The manuscript does not mention the values of the numerous parameters of the Reasenberg algorithm (the minimum look-ahead time for building clusters, the maximum look-ahead time for building clusters, a confidence probability, the effective lower cutoff magnitude, the increase in lower cutoff magnitude during clusters, and the number of crack radii surrounding each earthquake), while these parameters can have a profound impact on the results of the declustering; 2) The manuscript seems to ignore the earthquakes occurring outside the study area or previously to 2018. This is problematic as these un-considered earthquakes have local magnitude up to 5, while the largest earthquake in the space-time window considered is only a magnitude of 1.4 (which is extremely low). It is unclear to me that there is no statistical link between what is observed in the study area and what occurred before or outside. The manuscript needs to better address this issue and demonstrate that there is no link.

- Computation of pore pressure and the issue of time variation: The manuscript uses (in its main text) only one value of the coefficient of pore pressure diffusion D but evaluates the DCFS obtained when changing this coefficient D in the supplementary. On Fig 2 c, one can indeed observe that the modeled signal of pore pressure change, with $D=0.3 \text{ m}^2/\text{s}$ could be a good candidate for explaining the observed change in earthquake frequency. However, not only does the amplitude of the pore pressure signal changes with D , but the diffusion timescale also as it scales with L/\sqrt{D} , with L being the distance between the pore pressure source and the observed change in pore pressure. And it is relatively well admitted that D is not well constrained and could vary by orders of magnitude (which the manuscript fairly mentions lines 206-210). This means that the timing of when pore pressure increases or peaks is directly determined by D . There is in my opinion, no clear reason to choose $D=0.3$, except that it gives a good match with the change in earthquake activity. But using this argument would lead to a circular argument, which must be avoided. I would like the authors to account for the issue of the timing of pore pressure change, and to show/discuss the impact of changing pore pressure through time. The manuscript also needs to clearly mention/explain to what temporal state their maps of DCFS or pore pressure refer to (I have not found this information in the manuscript).

- Heterogeneous value of the coefficient of pore pressure diffusion: Moreover, the value of D changes spatially, in particular in areas crossed by faults or structural discontinuities (which is the case of this studied area where a fault is almost below the lake). This means that the spatial and temporal response to the filling of the lake is probably not the one of a homogeneous medium. The authors do not mention

this possibility, nor discuss the implications of this heterogeneity.

- The issue of depth: As shown by Figure S6, most earthquakes possibly triggered by the filling of the LDLs are located at a depth deeper than 5 km (about 6 km below the surface). This is puzzling, as the DCFS is likely to be significantly higher closer to the surface. Steer et al. (2014) or De Novellis et al. (2020, 2021) have clearly shown this dependency on depth. And some earthquakes can occur at shallower depths than 5-6 km. For instance, De Novellis et al. (2020) reports a depth of 1-2 km. So, my question is why the earthquakes you believe were triggered by filling of the LDLs do not occur at shallower depths, where the DCFS and pore pressure are likely to be of greater amplitude? This study need to make a clear investigation of the role of depth on DCFS and pore pressure (accounting for time as well), to show whether the earthquakes occur at an optimal depth, and if it is note the case, if other factors (such as the rheological nature of the faults – for instance the a and b parameters of the rate-and-state law) prevent earthquakes occurring at shallower depths than the one observed. A second question: what is the uncertainty of the depth estimate of these earthquakes? Generally, the depth is less well constrained than the horizontal position.
- Issue with the global study on the effect of LDL: the last part of the manuscript is dedicated to investigating how LDL could globally impact seismicity. However, I am not convinced by this part. First, the manuscript considers that all LDLs occur in a similar structural-tectonic setting, dominated by strike-slip faults, while this is likely to be incorrect. What would be the effect of considering normal or reverse faults? The localization of the faults and the orientation of the plane also significantly matters, but this is not considered in the manuscript. This study really needs to go deeper into this issue, or otherwise remove it as it is currently unconvincing.
- The paper correctly mentions that the geometry of the faults (except that they are likely in strike-slip regimes) are unknow. This study needs either to test the sensitivity of the DCFS results to the geometry at depth of the faults or to propagate the uncertainty on the geometry of these faults on the obtained values of DCFS.

Line comments:

- Lines 70-71 “deposited 2.5×10^7 m³ of sediments intor the river” – How do you know?
- Line 72 “upstream inflow of 1,680 m³/s” – Same, how do you know?
- Lines 72-75 – How were all these measurements obtained? Gauging stations, geotechnical reports, ... ?
- Line 83 “National Earthquake Data Center” – of which country?
- Lines 108-109: It is generally recommended to use more than 100 earthquakes to characterize mc and b.
- Lines 206-209: “S5). In addition, since crustal hydraulic diffusivity has been found to range from 0.01 to 5 m²/s (ref.45,46),we conservatively assume a diffusion coefficient of 0.3 m²/s” – In this case this is not a conservative estimate – please correct.
- Lines 226-227: “Wenchuan earthquake triggered more than 10,000 landslides” - Available landslide catalogs point towards more than 100,000 landslides and near 200,000 (e.g, Xu et al., 2014). Of course there is a large uncertainty, but still, more than 10,000 does not make justice to the estimated number of landslides.
- Lines 229-232: Reading these two sentences can wrongly give the impression to the reader that these numerous post-Wenchuan earthquakes were triggered by LDLs, which is obviously wrong. Of course, there are some aftershocks after a Mw7.9 mainshock. The effect of LDLs, if demonstrated, is likely to be negligible by orders of magnitude.

- Line 240-241: “Since landslide materials are denser than water, landslides smaller than our LDLs can result in comparable DCFS.” - Since landslides do not represent a new source of mass (but only the very local displacement of material, along the landslide path), this statement is likely wrong (except for very shallow faults of a few 100 m depths at more). The removal by erosion and sediment transport could play a role (Steer et al., 2020; Jeandet-Ribes et al., 2020), but this is different from what is mentioned here.

Reviewer #3 (Remarks to the Author):

This manuscript aims to understand the potential link between the Baige landslide dam and triggered small earthquakes in this region. It's intriguing to explore the interaction between earthquakes and surface hazards. The Baige landslide dam may indeed trigger some small earthquakes, but the current evidence is insufficient or inconclusive. Firstly, the observation period in 2018-2022 is relatively short and the network coverage is not shown in this study. Secondly, the limited number of events (39 with $ML \leq 1.4$) near the LDLs may not be representative of the earthquake activity in the study region. Thirdly, the estimate of M_c for 0.3 may not be reliable from the frequency-magnitude distribution of events with ML mostly less than 1 and a limited number of earthquakes (< 50). The locations of potential triggered earthquakes seem to be best correlated with the area with positive stress caused by the pore pressure changes than the stress due to the change of surface loads. It remains unclear whether fluids can infiltrate to the median hypocentral depth of 6 km within ~ 10 days. My other comments are listed below

Lines 69-71: To give a better overview of the Baige landslide dam, authors can provide a few photos of landslide-dammed lakes (e.g. photos shown in Zhang et al. Landslides, 2019) in the supplementary materials. It is not easy to follow the description of texts in Lines 70- 101 without a good figure. How landslide materials distribute in space during the first and second landslides and where the LSDs are located during two sliding events?

Lines 104-109: Please plot the location of seismic stations in Figure 1. The Baige landslide occurred on the Tibetan plateau where the station coverage may be insufficient to have the magnitude of completeness (M_c) of 0.3. In addition, most events have magnitudes ranging from 0 to 1 (Fig. 1b), is the value of M_c a robust estimate? Given only 48 events in May 2018 – Dec. 2022 and 36 events left after performing declustering, the temporal pattern of seismicity may be questionable.

Lines 113-114 The authors need to provide a detailed figure showing the time series of earthquake number and water level (daily sampling?) in the week of 12 to 18 November after the second LDL approached its peak water level. Why they choose earthquakes within 15 km of the LDLs but not a larger or smaller area? Are those chosen earthquakes shown as green dots in Fig. 2? Figure 1 shows earthquakes in this region and the seismicity activity seems to be high to the southeast of the study area with a subtle influence of LDLs. If authors choose earthquakes in different regions, how figure 2b varies accordingly?

Line 127-137, Please shows the locations of earthquakes before and after performing declustering. After declustering, where and how deep are the remaining eight earthquakes located? As authors claimed the largest magnitude (ML) is 1.4 during their observation period, maybe declustering is not necessary. Since there are only eight events (within 15 km of LDLs) left after declustering, I'm not sure the statistic test for the significance of increasing seismicity rate is meaningful given very few earthquakes in this period.

Lines 140-148: Since most earthquakes occurred after the second LDL are very small ($ML < 1.4$), it is unclear whether these earthquakes truly occurred on the surrounding fault system. Is the Coulomb stress estimated at the depth of 6 km in Fig. 2a? If so, please add it in the figure caption.

Lines 149-159: Please add a new figure to show the earthquake number from Nov. 10th to 20th where most earthquakes occurred. The stress due to fluid pressure diffusion reached peak about 12-13 days after the formation of landslide dam on Nov. 3rd. However, most of the hypocentral depths range from 5 to 15 km (median at 6 km) and it is unclear whether fluids can infiltrate to such great depths within ~10 days. About 80% of the earthquakes locate within regions of positive ΔCFS (Fig. 2a). However, the ΔCFS in Fig. 2a is mainly contributed by fluid pressure diffusion instead of direct loading.

Line 179: What are the minimum and median heights of reservoirs?

Lines 180-181: Fig 3a show the height of LDLs which induced earthquakes are generally lower than the heights of reservoirs induced earthquakes. I don't understand the logic of "Therefore, a significant number of LDLs globally are of similar heights as reservoirs which induced earthquakes."

Line 286: How did authors obtain distributed surface forces?

Figure 1 is not very informative. Please mark the downstream direction. A more detailed map of LDLs.

Response to Reviewer #1:

- What are the noteworthy results?

This is generally a well written paper with a provocative suggestion, that landslides could potentially create a large enough of stress disturbance that will trigger earthquakes. This assertion is not far-fetched, as we see many are documenting induced seismicity throughout the world, from gas extractions, enhanced oil recovery, hydraulic fracturing, reservoir induced seismicity, and even seismicity from glacial retreat. However, this work depends heavily on the earthquake catalog that the authors did not produce, and I worry that all of uncertainties are not explored, making the case very weak. In its current state, I cannot recommend this paper for publication.

We would like to thank you for the specific and constructive comments you provided for this manuscript. These comments are really helpful and greatly improve this manuscript. We are honored that you are interested in our results and agree that a high-quality earthquake catalog is critical for strengthening our main conclusion that landslide hazard cascades can trigger earthquakes. Therefore, in the revised manuscript, based on the continuous seismic waveforms from January 2014 to January 2023, we use machine learning and cross-correlation-based methods to build an earthquake catalog which contains ~5 times more events than the previously presented catalog. We find that ~61 earthquakes of magnitudes up to 2.6 occurred in the week when the second landslide-dammed lake LDL was near peak water level, substantially strengthening our main novel conclusion that landslide hazard cascades can trigger earthquakes. Furthermore, we discuss in detail the uncertainties in earthquake location and stress calculations based on this high-quality earthquake catalog and confirm that our main conclusions still stand after considering these uncertainties. We believe that this revised manuscript has addressed your major concerns and look forward to receiving further feedback.

- Does the work support the conclusions and claims, or is additional evidence needed?

The previously existing seismic network was enhanced in May 2018, and because of this, the sensitivity of the network changed, allowing for more small magnitude earthquakes to be detected and located. Thus, it is difficult to determine the background seismicity rate prior to the landslide in Nov. 2018, and the seismicity rate of the region in general.

Thank you for your comments. We used cross-correlation and machine learning based methods to build an enhanced earthquake catalog from January 2014 to January 2023 which allows good estimation of the background seismicity rate of the region.

Before more seismic stations became operational in May 2018, no earthquakes were detected within 10 km of the LDLs due to limited station coverage and/or low seismicity rate. Nevertheless, in the ~250 weeks of observation period since May 2018 for which we have an improved earthquake catalog, 61 earthquakes occurred within 10 km of the LDLs during the week when the second LDL's water level almost peaked while there were only 183 earthquakes over the ~247 weeks outside the time period of the two LDLs i.e., a background seismicity rate of only ~0.7/week over ~5 years (Table S1). Therefore, our catalog shows that the seismicity rate during the LDL is significantly higher than the background seismicity rate of the region.

Furthermore, the authors do not plot the station distribution (before or after May 2018),

and the depth uncertainties are not stated. If these events are deep in the crust, making the case for induced seismicity becomes very difficult. Even if they were shallow, we have no idea of the station distribution and true depth uncertainly, plus the diffusivity should be known to make any connection to water migration and the timing of the proposed induced seismicity. To have good depth determination, a seismic station must be within a focal depth to have any chance of constraining depth, plus having a well calibrated velocity model is critical for accurate depth estimates. There is no mention of the station distribution, the local velocity model, the location errors, and the number of stations that recorded the events. Furthermore, these are catalog events, and perhaps the seismicity is clustered near the landslide itself. Using an approach like HypoDD (double difference) might show that the events are clustered spatially, invalidating the argument that the events are spread throughout the region of inundation. None of these location approaches are explored; thus, the correlation with the landslide is tenuous at best. Lastly, all of the triggered events seem to be less than $M_{L}1.4$. These are micro earthquakes, and they are difficult to locate and analyze. The seismology for these earthquakes will have to be more clearly documented for this work to be convincing.

Thank you for these comments. We now show the distribution of seismic stations in Fig. 1.

Fig. 1 | Study site and seismicity. a Distribution of seismic stations. Red box marks the region covered by the earthquake catalog we built. Black and green triangles indicate seismic stations that started operating since January 2014 and May 2018, respectively. **b Map of seismicity from January 2014 to January 2023.** Red beach balls depict focal mechanisms available from the National Earthquake Data Center of strike-slip earthquakes³⁹. Red and gray dots represent earthquakes which occurred when the second landslide-dammed lake (LDL) almost reached peak water level and earthquakes beyond that, respectively. Red box marks the region shown in (d). **c Magnitude-frequency distribution of earthquakes from May 2018 to January 2023**

*within 10 km of the LDLs. Magnitude of completeness (M_c) and a - and b -values were calculated using the ZMAP software⁴⁰. **d Distribution of earthquakes from May 2018 to January 2023 within 10 km of the LDLs.** Red and gray dots represent earthquakes that occurred when the second LDL almost reached peak water level and earthquakes beyond that, respectively.*

We also introduce the methods of earthquake detection, location, magnitude calculation, and discuss the location uncertainties in detail.

We adopt an AI-based workflow to develop a new earthquake catalog from 2014 to 2023 based on the 13 permanent seismic stations⁵⁷ (Fig. 1). We first use the machine-learning-based detector, PhaseNet⁵⁸, to identify P- and S-wave arrival times from the continuous waveforms. Subsequently, these P- and S-wave arrival times are associated into individual earthquakes using a high-throughput seismic phase associator, PyOcto⁵⁹, and a local 1-D velocity model⁶⁰. A threshold of two P picks, one S pick, and a total of five P and S picks is adopted during the PyOcto association, resulting in 6,692 earthquakes within a region of $2.5^\circ \times 3^\circ$ (latitude: 30° to 32.5° ; longitude: 97° to 100°). We then manually inspect the waveforms of 859 earthquakes around our target area (latitude: 30.7° to 31.7° ; longitude: 98.1° to 99.2°) to exclude 112 non-tectonic or unreasonably associated events. We then use an absolute location method Hypoinverse⁶¹ to further refine the hypocenters of the remaining 747 events and calculate their location uncertainties.

We also estimate the local magnitudes for all retained earthquakes based on their S-wave amplitudes and a recently improved national standard magnitude scale that is specific to the Sichuan region, China^{62,63}. The maximum amplitudes of horizontal component waveforms are measured after deconvolving the instrument response from the raw waveforms and then convolving the obtained signal with the theoretical Wood-Anderson seismometer response. The measured waveform window starts 0.5 seconds before the P wave arrival and is twice the predicted S–P travel time in length. Note that we only estimate magnitudes for 669 events with at least two stations having signal-noise ratios of S waves larger than 3. The estimated magnitudes range from M_L 0.1 to 4.2.

To further improve the magnitude of completeness, we use a template matching technique, Graphics Processing Unit-based match and locate technique (GPU-M&L)⁶⁴, to detect and locate more earthquakes based on the 669 retained earthquakes with estimated magnitude. P- and S-waves of these events are simultaneously used in GPU-M&L. The length of template window is 6 s starting 1 s before the P- and S-wave arrivals picked by PhaseNet. We filter the continuous and template waveforms from 2 to 8 Hz. The 1-D velocity model used in the above earthquake absolute location was adopted in GPU-M&L as well. Note that in GPU-M&L, we relocate the epicenters of detections through a grid search method with searching space $1.2 \text{ km} \times 1.2 \text{ km}$ in the horizontal direction and searching interval of 0.06 km if template and continuous waveforms were recorded by at least three seismic stations. Otherwise, the locations of detections will be defined as that of the templates. By manually inspecting these detections, the detection threshold is defined as correlation coefficient of 0.3 and 12

times median absolute deviation (MAD) if the common component number between template and continuous waveforms is ≥ 9 . Otherwise, the detection threshold is defined as correlation coefficient of 0.5 and 15 times MAD. In total, GPU-M&L resulted in 4,295 detections including the 669 template events. The magnitudes of the newly detected earthquakes are estimated based on the local magnitudes of the templates and the amplitude ratio between these detections and their corresponding templates⁶⁵. Note that due to the poor local seismic station coverage with available data before May 2018, 22 earthquakes that occurred from March 2014 to May 2018 in the routine catalog, which utilized data from the entire China, were not detected. Thus, these missing earthquakes are manually added to our earthquake catalog.

We also attempted to use HypoDD to relocate the seismic events we detected. However, due to the limited seismic station coverage, many outliers appear in the relocation results hence we decided not to include the HypoDD results in the manuscript. Nevertheless, our new enhanced earthquake catalog shows that the earthquakes during the landslide hazard cascades were closely related to the LDLs in space (Fig. 2a) and time (Figs. 2c and S2) after location and depth uncertainties are taken into account, which supports our main conclusions:

Fig. 2 | Earthquakes triggered by 2018 Baige landslide-dammed lake. a Spatial distribution of Coulomb stress change (ΔCFS) at depth of 4.5 km corresponding to -0.5 km asl on 13 November 2018 and earthquakes from 10 November to 16 November 2018 within 10 km of the landslide-dammed lakes (LDLs). ΔCFS is from the combined effect of direct gravitational loading and fluid pressure diffusion (Fig. 2d). **b** Cumulative number of earthquakes and weekly seismicity rate over a five-year period. Arrows mark the timings of the two 2018 Baige landslides. **c** Earthquake magnitudes and LDL water level over a fifty-day period. **d** Temporal evolution of ΔCFS and pore pressure at point P (Fig. 2a).

Fig. S2 | Daily seismicity rate, earthquake magnitudes, and LDL water level over a twenty-day period.

As the second LDL approached peak water level, ~50% and ~95% of the hypocenters in the earthquake sequence are at depths of <1.2 and <4.5 km, respectively (Fig. S8). However, the mean horizontal and vertical location uncertainties of our cataloged earthquakes are ~4 and ~8 km, respectively. To further constrain the likely depths of these events, we look at the larger earthquakes which are more accurately located since they are recorded by more stations. We find that the median depth of all $M_L > 1.1$ earthquakes during the second LDL is ~1.5 km with average depth uncertainty of ~6 km. Our modelling shows that ΔCFS down to a depth of 7 km can reach levels that trigger earthquakes (Fig. S11d), and the ΔCFS is larger at shallower depths (Fig. S11). Furthermore, ~60% of reservoirs with documented earthquake depths have induced earthquakes up to >8 km depth (Fig. S14). Therefore, we conclude that the earthquake sequence can be triggered by stress changes due to the LDL.

Fig. S8 | Earthquake depth over an eight-day period.

Fig. S11 | Coulomb stress change (ΔCFS) map on 13 Nov 2018. a-d ΔCFS , as in Figure 2, assuming different fault depths. e-h ΔCFS assuming different fault dip values.

Furthermore, we realize that the earthquake magnitudes used in our previous manuscript initially reported in the China Unified Earthquake Catalog were underestimated. For the same earthquakes, our estimated local magnitude is 0.8-1.2 larger than the previous magnitude and consistent with the final report of the China Earthquake Networks Center. Therefore, the triggered events have M_L up to 2.6.

- Are there any flaws in the data analysis, interpretation and conclusions? Do these prohibit publication or require revision?

Although not under the author's control, the number and magnitude of earthquakes are small, and running statistics on these small numbers is difficult. The authors decluster the catalog and use a couple of statistical methods to determine a change in seismicity rate. Complicating this, associating the events to the water inundation caused by landslide proves difficult given the time delay (two weeks) before the onset of seismicity. I would explore the use of other statistics used for dynamic triggering studies (works of Z. Peng, K. Pankow, R. Alfaro-Diaz, D. Kilb, E. Brodsky). If these events are indeed triggered, they would be classified as delayed triggering, which even is harder to show statistically. Finally, seismicity also appears widespread in the region, not necessarily on the faults. Again, this may be the result of poor locations. There is simply no way to assess given the information given in the study.

Thank you for your suggestion. We agree with your concerns about how to confirm the spatiotemporal evolution relationship between these triggered earthquakes and the second landslide hazard cascade. Our new earthquake catalog gives us the opportunity to better analyze the evolution of this earthquake sequence. Based on the enhanced earthquake catalog, we observed that *as the second LDL approached peak water level, the number and magnitude of earthquakes started increasing as the water level rises and then peaked together with the water level peak (Figs. 2 and S2). Subsequently, as the LDL water level decreased following the dam breach, the earthquake activity gradually decreased back to the background rate.*

Furthermore, following your suggestion, we now also quantify the change in seismicity rate using the improved statistic based on Poisson probability (Alfaro-Diaz et al.,

2020)⁴⁴ and an empirically derived statistic (Pankow and Kilb, 2020)⁴⁵ (Fig. S6) tests (Methods) which were developed to determine statistically significant changes in earthquake rate under small background rate:

We further evaluate the statistical significance of the observed seismicity rate change based on both an improved statistic based on Poisson probability⁴⁴ and an empirically derived statistic⁴⁵ which were developed to determine statistically significant changes in earthquake rate under small background rate. We assumed that earthquakes occur independently at a constant rate, following a Poissonian distribution⁴⁴. To determine statistical significance of seismicity rate increase, we compare the number of expected earthquakes in a 1-week window to the number of earthquakes in the 1-week window after the trigger, and calculate the Poisson probability of getting the number of earthquakes (v) in the 1-week window after the trigger given the expected number of events (μ) in the 1-week window.

$$P_{\mu}(v) = e^{-\mu} \frac{\mu^v}{v!} \quad (3)$$

For this case of low background seismicity rate, similar to the idea of Alfaro-Diaz et al. (2020)⁴⁴, we conservatively regard the largest number of earthquakes (3) in a 1-week window over our observation period, except for this earthquake sequence, as the expected number of events (μ) in a 1-week window. Based on Equation 3, we obtain a P value of 0.0027 taking μ and v as 3 and 9 respectively. This implies that our observed seismicity rate increase after declustering⁴¹ is statistically significant at a >99.73% level.

We also use empirical statistical method of Pankow and Kilb (2020)⁴⁵ to determine the significance in the seismicity rate change. Firstly, we count the number of earthquakes in each 1-week time window incorporating a 1-day sliding window. There are >1,700 windows from May 2018 to January 2023. We then assign the number of earthquakes N_{count} within each window to a timestamp at the start of the window. We next build a histogram of N_{count} values (Fig. S6). Finally, using percentile measurements, the >99% statistically significant level is 3 earthquakes in a 1-week window. We observed 9 earthquakes (after declustering) as the second LDL approached the peak water level, hence this seismicity rate increase is statistically significant at a >99% level.

Fig. S6 | Number of earthquakes in a given 1-week time window using the earthquakes from May 2018 to January 2023. The inset figure zooms into the smaller values of the histogram.

Regarding your concern of running statistics on small numbers, our enhanced catalog now contains more earthquakes and we also perform four different statistical tests (including one developed to determine statistically significant changes in earthquake rate under small background rate) using the whole, >Mc, and declustered catalogs, all of which confirm that the increase in seismicity rate is statistically significant at >99% level.

Table S1 | Seismicity rate change and statistical significance.

Time window	Seismicity rate per week		
	All events	>Mc events	Declustered events
1 week during the second LDL*	61	16	9
27 weeks before the second LDL (Total number of earthquakes)	0.7 (19)	0.3 (9)	0.3 (8)
224 weeks after the second LDL (Total number of earthquakes)	0.7 (164)	0.2 (37)	0.2 (37)
Maximum over observation period besides the second LDL	14	3	3
Statistical tests	Statistical significance		
	All events	>Mc events	Declustered events
Statistic P	100.00%	100.00%	99.82%
Statistic Z	100.00%	100.00%	99.62%
Improved statistic based on Poisson probability	100.00%	100.00%	99.73%
Empirically derived statistic	99.88%	99.71%	99.71%

* 1 week (10 to 16 November) when the second LDL approached its peak water level

Finally, the locations of the triggered earthquakes in our enhanced earthquake catalog are mainly distributed near previously mapped faults (Fig. 1). The triggering is also not delayed since the seismicity rate started increasing as the second LDL approached peak water level (Fig. 2c).

• Is the methodology sound? Does the work meet the expected standards in your field?

Stress modeling also depends on assumed faults. The fault information also appears not well documented (orientation of the faults are not known). Determining focal mechanism would help, but these are small earthquakes. Perhaps a composite focal mechanism can be used to determine what the stress direction is that would be released with increase pore pressure. Furthermore, how deep are these faults, and how long would it take for the water to reach these faults?

Thank you for your suggestion. The fault information is indeed limited because large earthquakes occur rarely in this region. Nevertheless, since January 2014, four $M > 3.5$ earthquakes occurred at 16 to 35 km from the LDLs and have similar left-lateral strike-slip focal mechanisms. While the small event sizes and limited station coverage prevent us from directly inverting for the focal mechanisms of the earthquake sequence during the LDL, we find that the first motions recorded by surrounding seismic stations of the two largest earthquakes as the second LDL approached peak water level are generally consistent with the left-lateral strike-slip focal mechanisms of the surrounding large earthquakes (Fig. S7). Finally, these focal mechanisms are also consistent with the

strike and fault type documented based on field investigations. Therefore, we believe our assumed fault geometry is reasonable.

Fig. S7 | Focal mechanism of M_L 3.7 earthquake and the first motions of two $M_L > 1.8$ earthquakes at their surrounding seismic stations. Red beach balls depict focal mechanisms of M_L 3.7 earthquake that occurred on 30 June, 2023 (striking: 323° , dip: 74° , slip: -16°) in Fig. 2b. For the earthquake sequence which occurred within 10 km of the LDL as the second LDL approached peak water level, the first motions of the two largest earthquakes (M_L 2.6 (a) and M_L 1.9 (b)) at surrounding seismic stations are marked. Black and white dots represent upward and downward motions, respectively.

Finally, in the revised manuscript we also discuss the effects of fault depth and dip on stress changes on surrounding faults and confirm that our main conclusions stand after considering these uncertainties.

The stress change on surrounding faults due to the LDLs partly depends on the fault geometry. In contrast to the well-documented fault strike, the dip of the fault is less well constrained. We assume a fault dip of 75° NE based on the focal mechanisms of four $M > 3.5$ earthquakes nearby. In addition, our modelling shows that the fault dip mainly affects the spatial distribution and amplitude of Δ CFS but not its sign, and the seismicity generally locate within regions of positive Δ CFS for a range of different assumed fault dips (Fig. S11e-h). Hence, the dip uncertainty has minimal effect on our conclusion that landslide hazard cascades can trigger earthquakes.

Response to Reviewer #2:

Zhang et al. focuses on the potential impact of landslide-dammed lakes on the triggering of earthquakes on nearby faults. The paper is easy to read and addresses an interesting issue, which I feel deserves to receive the attention of a high-impact journal. Too few studies have indeed focused on the impact of surface processes during extreme events on the triggering of earthquakes and deformation. I was therefore thrilled to receive this paper for a review.

Unfortunately, I have found too many major weaknesses that at this stage prevent, in my opinion, the paper from being published. Despite that, I also feel that most of these major issues can be thoroughly addressed by the authors, and I therefore would ask the editor and authors to consider a phase of major revisions, but not necessarily of rejection (or otherwise with the opportunity for resubmission). I want to thank the authors for this interesting manuscript, and I hope they will receive most of these comments as a means to increase the quality of their manuscript, and not only as a criticism. I first summarize below my main comments and then describe my other comments line-by-line.

With my best regards,

Philippe Steer

We would like to thank you for your very constructive comments on our manuscript. They have helped to significantly improve this manuscript in all aspects. We have revised the manuscript according to your great suggestions and comments.

Main comments:

- False claim of the manuscript: The manuscript repeats several times (lines 14-15, 49-52, in conclusion as well) that this is the first time that a link between surface hazards/processes (or erosion or landslides) and the triggering of earthquakes is made. This is simply wrong and surprising to see this affirmation, as several visible papers in high-impact journals were published on this subject during the last 15 years: Calais et al. (2010 - Nature), Steer et al. (2014 - Nature Communications), Steer et al. (2020 - Scientific reports), Jeandet Ribes et al. (2020 - GRL), and probably others. Moreover, several papers have already shown that lake water variation can impact seismicity and trigger earthquakes (Gough & Gough, 1970; Kebeasy & Gharib, 1991; Simpson et al., 2018; and many more). This paper is therefore not the first one to address the issue of the triggering of earthquakes because of surface hazards, landslides, or erosion, nor it is the first one to show the impact of lake water level fluctuation on earthquake triggering. The manuscript by Zhang et al. therefore needs to account for these previous studies, in particular in the title, introduction, discussion and conclusion, and to correct the importance of their study respectively to what was already demonstrated by these previous studies. This manuscript is however the first one (if I have not missed a previous article) to address the issue of the response of seismic activity to landslide-dammed lakes filling. But this is largely different from what the current manuscript claims.

We are very sorry that we have overlooked these important references. We have modified the relevant descriptions in our manuscript (including title, abstract and conclusion) to include these important references and communicate the novelty of our specific results while acknowledging the contributions of these previous studies.

• Issue with declustering: The manuscript uses two classical approaches to decluster the earthquake catalog they investigate. However, only 3 over 39 earthquakes are identified as aftershocks and removed from the catalog. This is a very unusually low number of earthquake removal by declustering. I identify two issues that need to be addressed by the manuscript: 1) The manuscript does not mention the values of the numerous parameters of the Reasenberg algorithm (the minimum look-ahead time for building clusters, the maximum look-ahead time for building clusters, a confidence probability, the effective lower cutoff magnitude, the increase in lower cutoff magnitude during clusters, and the number of crack radii surrounding each earthquake), while these parameters can have a profound impact on the results of the declustering; 2) The manuscript seems to ignore the earthquakes occurring outside the study area or previously to 2018. This is problematic has these un-considered earthquakes have local magnitude up to 5, while the largest earthquake in the space-time window considered is only a magnitude of 1.4 (which is extremely low). It is unclear to me that there is no statistical link between what is observed in the study area and what occurred before or outside. The manuscript needs to better address this issue and demonstrate that there is no link.

Thank you for your suggestion. Using continuous seismic data recorded by 13 nearby seismic stations from January 2014 to January 2023, we apply machine-learning and cross-correlation-based methods to build an enhanced earthquake catalog which contains 3,970 earthquakes with documented magnitude within a region of $\sim 1^\circ \times 1^\circ$ (Fig. 1b). 244 of these earthquakes within 10 km of the LDLs occurred after May 2018 when a denser seismic network began operation.

Fig. 1 | Study site and seismicity. a Distribution of seismic stations. Red box marks the region covered by the earthquake catalog we built. Black and green triangles indicate seismic stations that started operating since January 2014 and May 2018, respectively. **b Map of seismicity from January 2014 to January 2023.** Red beach balls depict focal mechanisms available from the National Earthquake Data Center of strike-slip earthquakes³⁹. Red and gray dots represent earthquakes which occurred when the second landslide-dammed lake (LDL) almost reached peak water level and earthquakes

beyond that, respectively. Red box marks the region shown in (d). **c Magnitude-frequency distribution of earthquakes from May 2018 to January 2023 within 10 km of the LDLs.** Magnitude of completeness (M_c) and a - and b -values were calculated using the ZMAP software⁴⁰. **d Distribution of earthquakes from May 2018 to January 2023 within 10 km of the LDLs.** Red and gray dots represent earthquakes that occurred when the second LDL almost reached peak water level and earthquakes beyond that, respectively.

To estimate the earthquake activity in our study area, we estimate the magnitude of completeness (M_c) to be 1.1 (ref.⁴⁰; Fig. 1c) for the earthquakes within 10 km of the LDLs. There are 62 earthquakes between May 2018 and January 2023 with local magnitude $M_L \geq 1.1$, 16 of which occurred in the week when the second LDL approached its peak water (Table S1). We then perform declustering using the Reasenber method⁴¹, leaving us with 54 events (Fig S5), 9 of which occurred in the week the second LDL approached its peak water level (Fig. S5, Table S1). For your two specific questions, we have made the following revisions:

Fig. S5 | Earthquakes after performing declustering. **a** Distribution of earthquake from May 2018 to January 2023 within 10 km from LDLs. Red and blue dots represent the locations of earthquakes that were left and removed after performing declustering, respectively. **b** Earthquake magnitudes and LDL water level over a twenty-day period. **c** Earthquake depths over a twenty-day period.

- 1) Values of the numerous parameters of the Reasenberg algorithm: *Using the approach from Reasenberg⁴¹, we identified 5 clusters of earthquakes containing a total of 13 events (5 main earthquakes and 8 foreshocks/aftershocks) out of 62 events which occurred over ~5 years, using the following input parameters⁴¹: look-ahead time from 0.5 to 3 days, confidence probability of 0.9, effective min magnitude cutoff of 1.6, the increase in lower cutoff magnitude during clusters of 0, and the number of crack radii surrounding each earthquake of 5. Most events are not identified as aftershocks which makes sense since the largest events are small ($M_L < 2.7$) and the events are spaced out in time (event rate of ~0.7/week).*
- 2) Statistical link between what is observed in the study area and what occurred before or outside: *We further verified that only 5 earthquakes (all with $M_L < 2.0$) occurred within 60 km of the LDLs besides this earthquake sequence as the second LDL approached peak water level (Fig. 1b), and the largest earthquake that occurred*

within 60 km of the LDLs in the six months before this earthquake sequence has M_L 2.7 (Fig. S4), hence the earthquake sequence within 10 km of the LDLs is unlikely to have been triggered by surrounding large earthquakes.

Fig. S4 | Distribution of $M_L > 2.5$ earthquake from May 2018 to November 2018. Time and magnitude of the earthquakes are marked.

• Computation of pore pressure and the issue of time variation: The manuscript uses (in its main text) only one value of the coefficient of pore pressure diffusion D but evaluates the DCFS obtained when changing this coefficient D in the supplementary. On Fig 2 c, one can indeed observe that the modeled signal of pore pressure change, with $D=0.3$ m^2/s could be a good candidate for explaining the observed change in earthquake frequency. However, not only does the amplitude of the pore pressure signal changes with D , but the diffusion timescale also as it scales with L/\sqrt{D} , with L being the distance between the pore pressure source and the observed change in pore pressure. And it is relatively well admitted that D is not well constrained and could vary by orders of magnitude (which the manuscript fairly mentions lines 206-210). This means that the timing of when pore pressure increases or peaks is directly determined by D . There is in my opinion, no clear reason to choose $D=0.3$, except that it gives a good match with the change in earthquake activity. But using this argument would lead to a circular argument, which must be avoided. I would like the authors to account for the issue of the timing of pore pressure change, and to show/discuss the impact of changing pore pressure through time. The manuscript also needs to clearly mention/explain to what temporal state their maps of DCFS or pore pressure refer to (I have not found this information in the manuscript).

Thanks for your comments and suggestions. We have added Fig. S16 to show the temporal evolution of pore pressure and the ΔCFS in the revised manuscript. *We further confirm that the temporal evolutions of ΔCFS at a given point are similar for different hydraulic diffusivity, and that the peak seismicity rate coincides with the peak amplitude of ΔCFS assuming different hydraulic diffusivity (Fig. S16).*

Fig. S16 | Coulomb stress change (ΔCFS) and earthquakes triggered by 2018 Baige landslide-dammed lake. a Earthquake magnitudes and LDL water level over a fifty-day period. **Temporal evolution of ΔCFS (b) and pore pressure (c) at point P (Fig. 2a) assuming different hydraulic diffusivity coefficient values.**

Furthermore, we marked the date of ΔCFS in Figs. 2a and S15 in the legend: 13 November 2018, corresponding with peak water level of the second LDL.

Fig. S15 | Coulomb stress change (ΔCFS) on 13 November 2018. a-d ΔCFS , as in Figure 2, assuming different friction coefficient values. e-h ΔCFS assuming different hydraulic diffusivity coefficient values.

Finally, we have added some description about the timing of pore pressure change and the temporal state of ΔCFS in the revised manuscript.

...The ΔCFS due to the first, smaller, and shorter-duration LDL peaked at 0.007 MPa which is smaller than the stress increase of >0.01 MPa typically associated with stress triggering of earthquakes^{47,48}. ...In comparison, about three weeks later on 3 November, the second landslide formed a larger LDL. The ΔCFS due to this second LDL peaked at 0.024 MPa when the LDL's water level approached its peak, coinciding with maximum earthquake activity (Fig. 2)....

• Heterogeneous value of the coefficient of pore pressure diffusion: Moreover, the value of D changes spatially, in particular in areas crossed by faults or structural discontinuities (which is the case of this studied area where a fault is almost below the lake). This means that the spatial and temporal response to the filling of the lake is probably not the one of a homogeneous medium. The authors do not mention this possibility, nor discuss the implications of this heterogeneity.

According to your suggestion, we have added some description about the heterogeneity:

The stress change on surrounding faults caused by LDL is also related to the fault friction coefficient and the fluid diffusion coefficient. We assumed a friction coefficient of 0.5 for our ΔCFS modelling. However, the chosen friction coefficient has minimal effect on both the spatial distribution and amplitude of ΔCFS (Fig. S15). In addition, due to differences in rock properties and depth-dependent confining pressure, crustal hydraulic diffusivity has been found to range from 0.01 to 5 m²/s (ref.^{67,68}). Both the geometry and closure of faults and fractures also cause significant spatial variation in hydraulic diffusivity^{69,70}. These factors make it challenging to obtain a precisely-constrained hydraulic diffusivity for the calculation of ΔCFS . Nevertheless, higher fluid diffusivity causes the pore water pressure to increase faster, which increases the amplitude of ΔCFS within a wider area. Our chosen value of 0.3 m²/s is on the lower end of observed values and is a typical value observed in fluid-driven seismic swarms in different regions^{71,72,73}.

• The issue of depth: As shown by Figure S6, most earthquakes possibly triggered by the filling of the LDLs are located at a depth deeper than 5 km (about 6 km below the surface). This is puzzling, as the DCFS is likely to be significantly higher closer to the surface. Steer et al. (2014) or De Novellis et al. (2020, 2021) have clearly shown this dependency on depth. And some earthquakes can occur at shallower depths than 5-6 km. For instance, De Novellis et al. (2020) reports a depth of 1-2 km. So, my question is why the earthquakes you believe were triggered by filling of the LDLs do not occur at shallower depths, where the DCFS and pore pressure are likely to be of greater amplitude? This study need to make a clear investigation of the role of depth on DCFS and pore pressure (accounting for time as well), to show whether the earthquakes occur at an optimal depth, and if it is not the case, if other factors (such as the rheological nature of the faults – for instance the a and b parameters of the rate-and-state law) prevent earthquakes occurring at shallower depths than the one observed. A second question: what is the uncertainty of the depth estimate of these earthquakes? Generally, the depth is less well constrained than the horizontal position.

Thanks for your comment. Based on the continuous seismic waveforms from January 2014 to January 2023, we use machine learning and cross-correlation-based methods to build an enhanced earthquake catalog. *As the second LDL approached peak water level, ~50% and ~95% of the hypocenters in the earthquake sequence are at depths of*

<1.2 and <4.5 km, respectively (Fig. S8). However, the mean horizontal and vertical location uncertainties of our cataloged earthquakes are ~ 4 and ~ 8 km, respectively. Therefore, the improved earthquake locations confirm that these triggered earthquakes are indeed located at shallow depths.

Fig. S8 | Earthquake depths over an eight-day period.

Furthermore, we perform a detailed analysis of the location uncertainties and confirm that these earthquakes occurred within a depth range where LDL filling can trigger earthquakes after accounting for the depth uncertainties. *To further constrain the likely depths of these events, we look at the larger earthquakes which are more accurately located since they are recorded by more stations. We find that the median depth of all $M_L > 1.1$ earthquakes during the second LDL is ~ 1.5 km with average depth uncertainty of ~ 6 km. Our modelling shows that ΔCFS down to a depth of 7 km can reach levels that trigger earthquakes (Fig. S11d), and the ΔCFS is larger at shallower depths (Fig. S11). Furthermore, $\sim 60\%$ of reservoirs with documented earthquake depths have induced earthquakes up to >8 km depth (Fig. S14). Therefore, we conclude that the earthquake sequence can be triggered by stress changes due to the LDL.*

Fig. S11 | Coulomb stress change (ΔCFS) map on 13 Nov 2018. a-d ΔCFS , as in Figure 2, assuming different fault depths. e-h ΔCFS assuming different fault dip values.

• Issue with the global study on the effect of LDL: the last part of the manuscript is dedicated to investigating how LDL could globally impact seismicity. However, I am not convinced by this part. First, the manuscript considers that all LDLs occur in a similar structural-tectonic setting, dominated by strike-slip faults, while this is likely to be incorrect. What would be the effect of considering normal or reverse faults? The

localization of the faults and the orientation of the plane also significantly matters, but this is not considered in the manuscript. This study really needs to go deeper into this issue, or otherwise remove it as it is currently unconvincing.

Thanks for your comment. To demonstrate that these LDL triggered earthquakes we shown are not special cases, we statistically analyzed global LDLs' characteristics, such as duration and dam height. Compared to reservoirs which induced earthquakes (Figs. 3a and 3b), we find that some LDLs also *have the potential to trigger earthquakes*.

Fig. 3 | General properties of landslide-dammed lakes. a Frequency distribution and cumulative distribution function of the height of reservoirs documented to have induced earthquakes and landslide-dammed lakes worldwide. **b** Frequency distribution and cumulative distribution function of the duration of landslide-dammed lakes worldwide. **c** Temporal evolution of ΔCFS and pore pressure at point P (Fig. 3d) and water surface elevation for a 30-m deep LDL. **d** Spatial distribution of ΔCFS at depth of 4.5 km corresponding to -0.5 km asl, 11 days after the formation of a 30-m deep LDL. ΔCFS is from the combined effect of direct gravitational loading and fluid pressure diffusion.

Furthermore, we acknowledge that it is unreasonable to use a similar structural-tectonic setting to calculate ΔCFS . Therefore, based on your suggestion, we state that *the stress responses to this typical LDL at different sites can vary (Fig. 3d) and depend on factors such as the fault types⁵¹ (Fig. S10), geometries (Fig. S11), and depths (Fig. S11). Nevertheless, our modelling shows that both the fault types and geometries mainly affect stress distribution rather than amplitude of ΔCFS (Figs. S10 and S11), and stress changes caused by the LDLs are larger at shallower depth (Fig. S11).*

Fig. S10 | Coulomb stress change (ΔCFS) map on 13 Nov 2018. a and b ΔCFS on the surrounding normal and reverse fault systems due to the LDL's direct gravitational loading and fluid pressure diffusion.

Finally, based on the characteristics of global LDL and calculated ΔCFS in a similar

structural-tectonic setting, we only state in the revised manuscript that other LDLs also have the potential to trigger earthquakes rather than give unconvincing specific number of LDLs which can trigger earthquakes. *Considering ~64% LDLs of the ~145 LDLs in the global database with documented duration lasted more than 11 days before dam breach (Fig. 3b) with no correlation with dam height (Fig. S12), our modelling suggests that other LDLs globally have the potential to trigger earthquakes if there are critically stressed faults nearby.*

- The paper correctly mentions that the geometry of the faults (except that they are likely in strike-slip regimes) are unknown. This study needs either to test the sensitivity of the DCFS results to the geometry at depth of the faults or to propagate the uncertainty on the geometry of these faults on the obtained values of DCFS.

Based on your suggestion, we added a test for the sensitivity of ΔCFS to both depth and dip of faults.

Our modelling shows that ΔCFS down to a depth of 7 km can reach levels that trigger earthquakes (Fig. S11d), and the ΔCFS is larger at shallower depths (Fig. S11).

The stress change on surrounding faults due to the LDLs partly depends on the fault geometry. In contrast to the well-documented fault strike, the dip of the fault is less well constrained. We assume a fault dip of 75°NE based on the focal mechanisms of four $M > 3.5$ earthquakes nearby. In addition, our modelling shows that the fault dip mainly affects the spatial distribution and amplitude of ΔCFS but not its sign, and the seismicity generally locate within regions of positive ΔCFS for a range of different assumed fault dips (Fig. S11e-h). Hence, the dip uncertainty has minimal effect on our conclusion that landslide hazard cascades can trigger earthquakes.

Line comments:

- Lines 70-71 “deposited 2.5×10^7 m³ of sediments into the river” – How do you know?

The two landslide hazard cascades are very well documented. There are many studies on these two events that present these field measurements in detail. We have added these references in the revised manuscript.

- Line 72 “upstream inflow of 1,680 m³/s” – Same, how do you know?

Similarly, we have added these references in the revised manuscript.

- Lines 72-75 – How were all these measurements obtained? Gauging stations, geotechnical reports, ... ?

Similarly, we have added these references in the revised manuscript.

- Line 83 “National Earthquake Data Center” – of which country?

We have removed this sentence in the revised manuscript because we built an enhanced earthquake catalog by ourselves.

- Lines 108-109: It is generally recommended to use more than 100 earthquakes to characterize m_c and b .

Since an enhanced earthquake catalog was built, we now have more than 100 earthquakes to evaluate M_c and b values in the revised manuscript.

• Lines 206-209: “S5). In addition, since crustal hydraulic diffusivity has been found to range from 0.01 to 5 m²/s (ref.45,46), we conservatively assume a diffusion coefficient of 0.3 m²/s” – In this case this is not a conservative estimate – please correct.

Done. *Our chosen value of 0.3 m²/s is on the lower end of observed values and is a typical value observed in fluid-driven seismic swarms in different regions^{71,72,73}.*

• Lines 226-227: “Wenchuan earthquake triggered more than 10,000 landslides” – Available landslide catalogs point towards more than 100,000 landslides and near 200,000 (e.g, Xu et al., 2014). Of course there is a large uncertainty, but still, more than 10,000 does not make justice to the estimated number of landslides.

Done. *...the 2008 M7.9 Wenchuan earthquake triggered more than 100,000 landslides and formed a few hundred LDLs^{52,53,54}*

• Lines 229-232: Reading these two sentences can wrongly give the impression to the reader that these numerous post-Wenchuan earthquakes were triggered by LDLs, which is obviously wrong. Of course, there are some aftershocks after a Mw7.9 mainshock. The effect of LDLs, if demonstrated, is likely to be negligible by orders of magnitude.

To avoid misleading readers, we have removed this sentence describing the aftershocks.

• Line 240-241: “Since landslide materials are denser than water, landslides smaller than our LDLs can result in comparable DCFS.” - Since landslides do not represent a new source of mass (but only the very local displacement of material, along the landslide path), this statement is likely wrong (except for very shallow faults of a few 100 m depths at more). The removal by erosion and sediment transport could play a role (Steer et al., 2020; Jeandet-Ribes et al., 2020), but this is different from what is mentioned here.

Thanks for your comments, we have removed these descriptions in the revised manuscript.

Response to Reviewer #3:

This manuscript aims to understand the potential link between the Baige landslide dam and triggered small earthquakes in this region. It's intriguing to explore the interaction between earthquakes and surface hazards. The Baige landslide dam may indeed trigger some small earthquakes, but the current evidence is insufficient or inconclusive. Firstly, the observation period in 2018-2022 is relatively short and the network coverage is not shown in this study. Secondly, the limited number of events (39 with $ML \leq 1.4$) near the LDLs may not be representative of the earthquake activity in the study region. Thirdly, the estimate of M_c for 0.3 may not be reliable from the frequency-magnitude distribution of events with ML mostly less than 1 and a limited number of earthquakes (< 50). The locations of potential triggered earthquakes seem to be best correlated with the area with positive stress caused by the pore pressure changes than the stress due to the change of surface loads. It remains unclear whether fluids can infiltrate to the median hypocentral depth of 6 km within ~ 10 days. My other comments are listed below

We would like to thank you for the specific and constructive comments you provided for this manuscript. We have studied these valuable comments carefully and tried our best to revise the manuscript according to these comments. Below is our point-to-point response to your concerns:

Firstly, regarding the issue of relatively short observation period in 2018-2022: Based on machine-learning and cross-correlation-based methods, we build an enhanced earthquake catalog based on the continuous waveforms from January 2014 to January 2023 recorded by 13 permanent seismic stations within ~ 200 km of the LDLs, of which 8 of these seismic stations have been operational since May 2018. Furthermore, we show the distribution of seismic stations in Fig. 1

Fig. 1 | Study site and seismicity. a Distribution of seismic stations. Red box marks the region covered by the earthquake catalog we built. Black and green triangles indicate seismic stations that started operating since January 2014 and May 2018, respectively. **b Map of seismicity from January 2014 to January 2023.** Red beach balls depict focal mechanisms available from the National Earthquake Data Center of strike-slip earthquakes³⁹. Red and gray dots represent earthquakes which occurred when the

second landslide-dammed lake (LDL) almost reached peak water level and earthquakes beyond that, respectively. Red box marks the region shown in (d). **c Magnitude-frequency distribution of earthquakes from May 2018 to January 2023 within 10 km of the LDLs.** Magnitude of completeness (M_c) and a - and b -values were calculated using the ZMAP software⁴⁰. **d Distribution of earthquakes from May 2018 to January 2023 within 10 km of the LDLs.** Red and gray dots represent earthquakes that occurred when the second LDL almost reached peak water level and earthquakes beyond that, respectively.

Before more seismic stations became operational in May 2018, no earthquakes were detected within 10 km of the LDLs due to limited station coverage and/or low seismicity rate. Nevertheless, in the ~250 weeks of observation period since May 2018 for which we have an improved earthquake catalog, 61 earthquakes occurred within 10 km of the LDLs during the week when the second LDL's water level almost peaked while there were only 183 earthquakes over the ~247 weeks outside the time period of the two LDLs i.e., a background seismicity rate of only ~0.7/week over ~5 years (Table S1). Therefore, our catalog shows that the seismicity rate during the LDL is significantly higher than the background seismicity rate of the region.

Secondly, regarding the limited number of events: Based on the continuous seismic waveforms from January 2014 to January 2023, we use machine learning and cross-correlation-based methods to build an enhanced earthquake catalog which contains ~5 times more events than the previously presented catalog. We find that ~61 earthquakes of magnitudes up to 2.6 occurred in the week when the second landslide-dammed lake LDL was near peak water level, substantially strengthening our main novel conclusion that landslide hazard cascades can trigger earthquakes.

Using continuous seismic data recorded by 13 nearby seismic stations from January 2014 to January 2023, we apply machine-learning and cross-correlation-based methods to build an earthquake catalog which contains 3,970 earthquakes with documented magnitude within a region of $\sim 1^\circ \times 1^\circ$ (Fig. 1b). 244 of these earthquakes within 10 km of the LDLs occurred after May 2018 when a denser seismic network began operation. It is well-established that earthquakes can trigger surface hazards and initiate hazard cascades^{5,6,7,12}. However, the Baige landslides were not triggered by earthquakes^{34,35,36,37,38} as no earthquakes with magnitude $M_L \geq 3.5$ were recorded within 50 km in the three preceding years (Figs. 1 and 2). Instead, in the week (10 to 16 November) when the second LDL approached its peak water level, 61 earthquakes with local magnitude up to ~2.6 occurred within 10 km of the LDLs. In comparison, there were only 19 earthquakes in the previous 27 weeks (~0.7 event/week) and 164 earthquakes in the subsequent ~224 weeks (~0.7 event/week) (Fig. 2, Table S1) in this region.

Thirdly, regarding the estimation of M_c given the small magnitude and limited number of events: Since an enhanced earthquake catalog was built, we now have enough earthquakes to better evaluate M_c and b values (Fig. 1c). Furthermore, we realize that the earthquake magnitude used in our previous manuscript initially reported in the China Unified Earthquake Catalog was underestimated. For the same earthquakes, our estimated local magnitudes are 0.8-1.2 larger than the previous magnitudes and consistent with the final report of the China Earthquake Networks Center. Therefore, the triggered events have M_L up to 2.6.

Fourthly, the issue of location: As the second LDL approached peak water level, ~50% and ~95% of the hypocenters in the earthquake sequence are at depths of <1.2 and <4.5 km, respectively (Fig. S8). However, the mean horizontal and vertical location uncertainties of our cataloged earthquakes are ~4 and ~8 km, respectively. To further constrain the likely depths of these events, we look at the larger earthquakes which are more accurately located since they are recorded by more stations. We find that the median depth of all $M_L > 1.1$ earthquakes during the second LDL is ~1.5 km with average depth uncertainty of ~6 km. Our modelling shows that ΔCFS down to a depth of 7 km can reach levels that trigger earthquakes (Fig. S11d), and the ΔCFS is larger at shallower depths (Fig. S11). Furthermore, ~60% of reservoirs with documented earthquake depths have induced earthquakes up to >8 km depth (Fig. S14). Therefore, we conclude that the earthquake sequence can be triggered by stress changes due to the LDL.

Fig. S8 | Earthquake depth over an eight-day period.

Fig. S11 | Coulomb stress change (ΔCFS) map on 13 Nov 2018. a-d ΔCFS , as in Figure 2, assuming different fault depths. e-h ΔCFS assuming different fault dip values.

Lines 69-71: To give a better overview of the Baige landslide dam, authors can provide a few photos of landslide-dammed lakes (e.g. photos shown in Zhang et al. Landslides, 2019) in the supplementary materials. It is not easy to follow the description of texts in Lines 70-101 without a good figure. How landslide materials distribute in space during the first and second landslides and where the LSDs are located during two sliding events?

Thank you for your suggestion. We have added some satellite images of these landslides in Fig. S1, and marked the spatial distribution of landslides, landslide dams and LDLs.

Fig. S1 | Sentinel-2 images near the Baige landslides. Dashed yellow and red lines outline the landslides and dams in b, c, and d. Time of these images are shown using yyyy-mm-dd format in each panel.

Lines 104-109: Please plot the location of seismic stations in Figure 1. The Baige landslide occurred on the Tibetan plateau where the station coverage may be insufficient to have the magnitude of completeness (M_c) of 0.3. In addition, most events have magnitudes ranging from 0 to 1 (Fig. 1b), is the value of M_c a robust estimate? Given only 48 events in May 2018 – Dec. 2022 and 36 events left after performing declustering, the temporal pattern of seismicity may be questionable.

Firstly, we plotted the locations of the seismic stations in Fig. 1. Furthermore, as you have seen, based on the continuous seismic waves from January 2014 to January 2023 recorded by 13 seismic stations, we use machine-learning and cross-correlation-based methods to build an enhanced earthquake catalog which contains 3,970 earthquakes with documented magnitude within a region of $\sim 1^\circ \times 1^\circ$ (Fig. 2a). 244 of these earthquakes within 10 km of the LDLs occurred after May 2018 when a denser seismic network began operation.

Fig. 1 | Study site and seismicity. a Distribution of seismic stations. Red box marks the region covered by the earthquake catalog we built. Black and green triangles indicate

seismic stations that started operating since January 2014 and May 2018, respectively. **b** *Map of seismicity from January 2014 to January 2023.* Red beach balls depict focal mechanisms available from the National Earthquake Data Center of strike-slip earthquakes³⁹. Red and gray dots represent earthquakes which occurred when the second landslide-dammed lake (LDL) almost reached peak water level and earthquakes beyond that, respectively. Red box marks the region shown in (d). **c** *Magnitude-frequency distribution of earthquakes from May 2018 to January 2023 within 10 km of the LDLs.* Magnitude of completeness (M_c) and a - and b -values were calculated using the ZMAP software⁴⁰. **d** *Distribution of earthquakes from May 2018 to January 2023 within 10 km of the LDLs.* Red and gray dots represent earthquakes that occurred when the second LDL almost reached peak water level and earthquakes beyond that, respectively.

We estimate the magnitude of completeness (M_c) to be 1.1 (ref.⁴⁰; Fig. 1c) for the earthquakes within 10 km of the LDLs. There are 62 earthquakes between May 2018 and January 2023 with local magnitude $M_L \geq 1.1$, 16 of which occurred in the week when the second LDL approached its peak water (Table S1). We then perform declustering using the Reasenber method⁴¹, leaving us with 54 events (Fig S5), 9 of which occurred in the week the second LDL approached its peak water level (Fig. S5, Table S1)....

.... Using the approach from Reasenber⁴¹, we identified 5 clusters of earthquakes containing a total of 13 events (5 main earthquakes and 8 foreshocks/aftershocks) out of 62 events which occurred over ~5 years, using the following input parameters⁴¹: look-ahead time from 0.5 to 3 days, confidence probability of 0.9, effective min magnitude cutoff of 1.6, the increase in lower cutoff magnitude during clusters of 0, and the number of crack radii surrounding each earthquake of 5....

Lines 113-114 The authors need to provide a detailed figure showing the time series of earthquake number and water level (daily sampling?) in the week of 12 to 18 November after the second LDL approached its peak water level. Why they choose earthquakes within 15 km of the LDLs but not a larger or smaller area? Are those chosen earthquakes shown as green dots in Fig. 2? Figure 1 shows earthquakes in this region and the seismicity activity seems to be high to the southeast of the study area with a subtle influence of LDLs. If authors choose earthquakes in different regions, how figure 2b varies accordingly?

We appreciate your suggestions. We have plotted a figure showing the time series of earthquake number and water level (Fig. S2). Furthermore, we chose to focus on earthquakes that occurred within 10 km of LDLs since the region with stress increase of >0.01 MPa is always located within 10 km of the LDLs (Fig. 2a). Nevertheless, the earthquake activity within larger and smaller region shows a similar accelerated trend during the landslide hazard cascades (Fig. S13).

Fig. S2 | Daily seismicity rate, earthquake magnitudes, and LDL water level over a twenty-day period.

Fig. S13 | Cumulative numbers of earthquakes occurred within 8 km (a), 12 km (b), and 14 km (c) of LDLs over a five-year period. Arrows mark the timings of the two 2018 Baige landslides.

Line 127-137, Please shows the locations of earthquakes before and after performing declustering. After declustering, where and how deep are the remaining eight earthquakes located? As authors claimed the largest magnitude (ML) is 1.4 during their observation period, maybe declustering is not necessary. Since there are only eight events (within 15 km of LDLs) left after declustering, I'm not sure the statistic test for the significance of increasing seismicity rate is meaningful given very few earthquakes in this period.

We have shown the locations and depth of earthquakes after performing declustering in Fig. S5. For the new earthquake catalog, we estimate the magnitude of completeness (M_c) to be 1.1 (ref.⁴⁰; Fig. 1c) for the earthquakes within 10 km of the LDLs. There are 62 earthquakes between May 2018 and January 2023 with local magnitude $M_L \geq 1.1$, 16 of which occurred in the week when the second LDL approached its peak water (Table S1). We then perform declustering using the Reasenber method⁴¹, leaving us with 54 events (Fig S5), 9 of which occurred in the week the second LDL approached its peak water level (Fig. S5, Table S1).

Fig. S5 | Earthquakes after performing declustering. *a* Distribution of earthquake from May 2018 to January 2023 within 10 km from LDLs. Red and blue dots represent the locations of earthquakes that were left and removed after performing declustering, respectively. *b* Earthquake magnitudes and LDL water level over a twenty-day period. *c* Earthquake depths over a twenty-day period.

Furthermore, we confirm that the increase in seismicity rate is also statistically significant at a >99% level (Table S1) based on both the improved statistic based on Poisson probability⁴⁴ and an empirically derived statistic⁴⁵ (Fig. S6) tests (Methods) which were developed to determine statistically significant changes in earthquake rate under small background rate.

Fig. S6 | Number of earthquakes in a given 1-week time window using the earthquakes from May 2018 to January 2023. The inset figure zooms into the smaller values of the histogram.

Regarding your concern that the statistical significance of increasing seismicity rate is due to very few earthquakes: our enhanced catalog now contains more earthquakes and we also perform four different statistical tests (including one developed to determine statistically significant changes in earthquake rate under small background rate) using the whole, >M_c, and declustered catalogs, all of which confirm that the increase in seismicity rate is statistically significant at >99% level. Therefore, the statistical significance of our result is not an artefact of having very few events.

Table S1 | Seismicity rate change and statistical significance.

Time window	Seismicity rate per week		
	All events	>M _c events	Declustered events
1 week during the second LDL*	61	16	9

27 weeks before the second LDL (Total number of earthquakes)	0.7 (19)	0.3 (9)	0.3 (8)
224 weeks after the second LDL (Total number of earthquakes)	0.7 (164)	0.2 (37)	0.2 (37)
Maximum over observation period besides the second LDL	14	3	3
Statistical tests	Statistical significance		
	All events	>Mc events	Declustered events
Statistic P	100.00%	100.00%	99.82%
Statistic Z	100.00%	100.00%	99.62%
Improved statistic based on Poisson probability	100.00%	100.00%	99.73%
Empirically derived statistic	99.88%	99.71%	99.71%

* 1 week (10 to 16 November) when the second LDL approached its peak water level

Lines 140-148: Since most earthquakes occurred after the second LDL are very small ($M_L < 1.4$), it is unclear whether these earthquakes truly occurred on the surrounding fault system. Is the Coulomb stress estimated at the depth of 6 km in Fig. 2a? If so, please add it in the figure caption.

In the revised manuscript, based on the continuous seismic waveforms from January 2014 to January 2023, we use machine learning and cross-correlation-based methods to build an earthquake catalog which contains ~5 times more events than the previously presented catalog. Our new enhanced earthquake catalog shows that the earthquakes during the landslide hazard cascades were closely related to the LDLs in space (Fig. 1d) and time (Figs. 1d and S2) after location uncertainties are taken into account, which supports our main conclusions. Furthermore, we estimate location uncertainties using the Hypoinverse. *As the second LDL approached peak water level, ~50% and ~95% of the hypocenters in the earthquake sequence are at depths of <1.2 and <4.5 km, respectively (Fig. S8). However, the mean horizontal and vertical location uncertainties of our cataloged earthquakes are ~4 and ~8 km, respectively. To further constrain the likely depths of these events, we look at the larger earthquakes which are more accurately located since they are recorded by more stations. We find that the median depth of all $M_L > 1.1$ earthquakes during the second LDL is ~1.5 km with average depth uncertainty of ~6 km. Our modelling shows that ΔCFS down to a depth of 7 km can reach levels that trigger earthquakes (Fig. S11d), and the ΔCFS is larger at shallower depths (Fig. S11).*

Fig. S8 | Earthquake depth over an eight-day period.

Finally, the Coulomb stress estimated at 4.5 km depth which is deeper than 95% depth of our observed seismicity (Fig. S8) and corresponds to ~ -0.5 km asl because $\sim 80\%$ of our study area is at elevation between 4 and 5 km asl³⁵. We added the depth for calculating Coulomb stress in the figure caption (Figs 2 and 3).

Lines 149-159: Please add a new figure to show the earthquake number from Nov. 10th to 20th where most earthquakes occurred. The stress due to fluid pressure diffusion reached peak about 12-13 days after the formation of landslide dam on Nov. 3rd. However, most of the hypocentral depths range from 5 to 15 km (median at 6 km) and it is unclear whether fluids can infiltrate to such great depths within ~ 10 days. About 80% of the earthquakes locate within regions of positive Δ CFS (Fig. 2a). However, the Δ CFS in Fig. 2a is mainly contributed by fluid pressure diffusion instead of direct loading.

We show the daily number of earthquakes in Fig S2, and show that the largest number of earthquakes occurs on November 13, which corresponds to the peak water level of the second LDL.

Fig. S2 | Daily seismicity rate, earthquake magnitudes, and LDL water level over a twenty-day period.

Furthermore, as the second LDL approached peak water level, $\sim 50\%$ and $\sim 95\%$ of the hypocenters in the earthquake sequence are at depths of <1.2 and <4.5 km, respectively (Fig. S8). However, the mean horizontal and vertical location uncertainties of our cataloged earthquakes are ~ 4 and ~ 8 km, respectively. To further constrain the likely depths of these events, we look at the larger earthquakes which are more accurately located since they are recorded by more stations. We find that the median depth of all $M_L > 1.1$ earthquakes during the second LDL is ~ 1.5 km with average depth uncertainty of ~ 6 km. Our modelling shows that Δ CFS down to a depth of 7 km can reach levels that trigger earthquakes (Fig. S11d), and the Δ CFS is larger at shallower depths (Fig. S11). Furthermore, $\sim 60\%$ of reservoirs with documented earthquake depths have induced earthquakes up to >8 km depth (Fig. S14). Therefore, we conclude that the earthquake sequence can be triggered by stress changes due to the LDL.

Line 179: What are the minimum and median heights of reservoirs?

...we find that the minimum and median heights of ~ 190 reservoirs with documented height that induced earthquakes are ~ 13 and 110 m, respectively (Fig. 3a).

Lines 180-181: Fig 3a show the height of LDLs which induced earthquakes are

generally lower than the heights of reservoirs induced earthquakes. I don't understand the logic of "Therefore, a significant number of LDLs globally are of similar heights as reservoirs which induced earthquakes."

We revised some descriptions: *In comparison, from a global database of ~410 LDLs reported worldwide spanning the period 1900-2018 (ref.⁵⁰), we find that of the ~300 LDLs with documented dam height, ~73% and ~10% were higher than the minimum and median heights of reservoirs which induced earthquakes, respectively (Fig. 3a). Therefore, some LDLs globally are of similar heights as reservoirs which induced earthquakes.*

Line 286: How did authors obtain distributed surface forces?

We added some descriptions: *The distributed surface forces can be estimated based on the spatiotemporal evolution of the LDL's water level. To quantify the spatiotemporal evolution of the LDL's water level, the LDL is mapped as a series of square cells with a size of 200 by 200 m. The elevation of each square cell is estimated from the GDEM V3 30 m. Based on the water depth at the dam (Figs. 2 and 3) and the elevation of each cell, the water depth of each cell can then be estimated. Therefore, the distributed surface forces, the direct gravitational loading effect of each cell, can be calculated using the water depth of each cell.*

Figure 1 is not very informative. Please mark the downstream direction. A more detailed map of LDLs.

We completely revised Fig. 1 and marked the flow direction. We also added Fig. S1 to show the distribution of landslides, landslide dams, and LDLs in detail.

Fig. S1 | Sentinel-2 images near the Baige landslides. Dashed yellow and red lines outline the landslides and dams in b, c, and d. Time of these images are shown using yyyy-mm-dd format in each panel.

REVIEWER COMMENTS

Reviewer #1 (Remarks to the Author):

Overall, I commend the authors for addressing my concerns by conducting new analysis, specifically, locating earthquakes using the available data. The depth uncertainty is still a bit ambiguous because of the lack of exploration of velocity models, and the reporting of the errors is realistic. Thus, the analysis is much stronger. My only comment on this draft is in regards to clarifying the figure captions (see below).

Specific Comments.

Figure 1: Need to state what the orange line is. I assume it is where the water was impacted. I also do not understand the red/grey circle description. Grey circles are earthquakes that occurred before the peak LDL? Red after?

“Red beach balls depict focal mechanisms...”. Beach balls are not what many like to refer these to. I would reword: “Focal mechanisms (red/white circles)...” The same style can be used throughout the caption:

Here’s a suggested re-write of the Figure caption:

Fig. 1: Study site and seismicity. a) Distribution of seismic stations operating since January 2014 (black triangles) and May 2018 (green triangles). Red box marks the specific study region where we developed the earthquake catalog. b) Seismicity map from January 2014 to January 2023 showing strike-slip focal mechanisms (red/white circles) available from the National Earthquake Data Center plus earthquakes that occurred before (grey filled circles) and after (red filled circles) when the second landslide-dammed lake (LDL) almost reached peak water level. Red box marks the region shown in (d). c) Magnitude-frequency distribution of earthquakes from May 2018 to January 2023 within 10 km of the LDLs. Magnitude of completeness (M_c) and a - and b -values were calculated using the ZMAP software. d) Distribution of earthquakes from May 2018 to January 2023 within 10 km of the LDLs.

I would recommend re-writing all captions for clarity, as illustrated above.

Reviewer #2 (Remarks to the Author):

Dear authors,

I wish to thank you for the hard and thorough work you have made to answer my comments as well as

the comments of the other reviewers. The quality and robustness of the manuscript have both been significantly improved thanks to that. If the manuscript still relies on some debatable assumptions (as in most Earth Sciences paper), they are now clearly stated and well discussed or justified. Demonstrating the link between an LDL event and few potentially triggered earthquakes (of small magnitudes) is statistically difficult. Yet, the authors have made a real effort to strengthen this link by improving the quality of the catalog and by exploring in more details the parametrization of the processes (pore pressure diffusion and direct gravitational loading) which could trigger earthquakes in relation to LDL water level rise.

I therefore suggest the editor to accept the paper for publication after some minor revisions (see below).

Best regards,

Philippe Steer

Minor comments

- Abstract: please explicitly state (line 21) that the observed increase in seismicity rate after the LDL only applies to low magnitude earthquakes
- Concerning pore pressure and its temporal evolution: Thank you for having added Fig. S16. However, I disagree with the interpretation that “the temporal evolutions of ΔCFS at a given point are similar for different hydraulic diffusivity, and that the peak seismicity rate coincides with the peak amplitude of ΔCFS assuming different hydraulic diffusivity (Fig. S16).” (as mentioned in your rebuttal letter and paper). Theory of pore pressure diffusion demonstrates that the pore pressure front progressively propagates from a point source at a distance d that is function of time t and Diffusivity D as $d = \sqrt{4 \pi D t}$ [see for instance Shapiro et al., 1997]). Therefore, diffusivity strongly impacts the timing of the arrival of the peak of pore pressure, as well as its amplitude, beneath the LDL. Fig. S16 (b and c) is also clearly showing that. You could strengthen your manuscript by 1) mentioning the range of D are consistent with having a DCFS above >0.01 MPa, and 2) by mentioning the relative contribution to DCFS of gravitational loading and pore pressure for these different values of D .
- Declustering: the manuscript only considers the lower ranges of accepted parameters for declustering using the Reasenberg approach (see for instance Table 3 of van Stiphout et al., 2012 <http://dx.doi.org/10.5078/corssa-52382934>). Please make additional tests with standard parameters and with the one of the lower range.

Reviewer #3 (Remarks to the Author):

The revised manuscript represents a significant improvement over the previous version, particularly in terms of the earthquake catalog and statistical analysis. However, I still have two principal concerns regarding the triggering mechanisms and earthquake statistics. While the manuscript has made notable

improvements, addressing these concerns will enhance the clarity and credibility of the research findings.

Man comments:

(1) While the manuscript provides a more comprehensive earthquake catalog and statistics, the correlation between the spatiotemporal occurrence of earthquakes and pore pressure changes remains unclear. It is noteworthy that most events ($M_L > 1.1$) occurred near the peak of the water level. However, the manuscript does not sufficiently elucidate the role of pore pressure throughout the entire process. Further clarification on how pore pressure influences seismic activity is warranted to enhance the understanding of the triggering mechanisms.

(2) It appears that the authors evaluate the statistical significance of the observed seismicity rate change by including events lower than the completeness magnitude (M_c). This approach may introduce biases and potentially obscure the true significance of seismicity rate changes. Revisiting the statistical analysis methodology to ensure that only events above the completeness magnitude are considered would strengthen the robustness of the findings and their interpretation.

Other comments:

1. Lines 121-124: By reviewing Figure 2 and the estimation of the number of events per week, it appears that the authors have included events with magnitudes lower than the completeness magnitude (M_c). As such, discussions regarding the seismic rate without considering M_c may lack meaningful interpretation. Therefore, I suggest removing these sentences for clarity and accuracy.

Figure 2:

Plot the cumulative number of earthquakes with magnitudes greater than or equal to M_c (1.1) in Figure 2b. This adjustment will provide a clearer representation of the seismic activity relevant to the study's objectives.

In Figure 2c, mark the value of M_c to highlight its significance in the context of the seismicity analysis. Additionally, plot the cumulative number of events since November 3 and compare the temporal evolution with the increase in water level. This comparison will offer valuable insights into the correlation between seismic activity and water level fluctuations.

2. Lines 167-180: The manuscript discusses Coulomb failure stress changes in the context of triggered seismic events, which predominantly exhibit small magnitudes (M_L less than 2). It is important to note that small events may not necessarily share similar geometries with adjacent fault systems.

3. Line 179: I have observed that the hypocentral depths of many events in the dataset appear to be pinned at specified depths (Fig S8). Consequently, these events may not be suitable for estimating the median depth of triggered earthquakes.

4. Lines 181-184: It appears that the authors solely estimated the increase in water level following the LSL, without accounting for the weight of the landslide sediments ($8.7 \times 10^6 \text{ m}^3$ as mentioned in Lines 98-99). Considering the substantial mass of the landslide sediments, which could exert gravitational

loading, it is crucial to evaluate its potential contribution to induced seismicity alongside the rise in water level.

5. Lines 227-229: How to estimate the temporal evolution of water level rise after the formation of LDL?

6. Lines 182-183: The manuscript lacks clarity regarding how the authors evaluate the contributions from direct gravitational loading of LDL and fluid pressure diffusion to induced seismicity. Despite the reported rise in water level, there is no discernible spatiotemporal migration of earthquakes associated with this increase. Notably, most events cluster near the peak of the water level, raising questions about the precise mechanisms driving seismic activity. Given the absence of clear migration patterns and the limited magnitudes of events, it remains uncertain whether the observed seismicity results from direct gravitational loading, fluid pressure diffusion, or other factors.

Figure 1 Cayan or Green triangles? Gray and red dots are too big.

Figure 2 See my earlier comment

Response to Reviewer #1:

Overall, I commend the authors for addressing my concerns by conducting new analysis, specifically, locating earthquakes using the available data. The depth uncertainty is still a bit ambiguous because of the lack of exploration of velocity models, and the reporting of the errors is realistic. Thus, the analysis is much stronger. My only comment on this draft is in regards to clarifying the figure captions (see below).

We are honored that you recognized our efforts to address your concerns. We would like to thank you for your great comments on our manuscript. They have helped to improve this manuscript in all aspects.

Specific Comments.

Figure 1: Need to state what the orange line is. I assume it is where the water was impacted. I also do not understand the red/grey circle description. Grey circles are earthquakes that occurred before the peak LDL? Red after? “Red beach balls depict focal mechanisms...”. Beach balls are not what many like to refer these to. I would reword: “Focal mechanisms (red/white circles)...” The same style can be used throughout the caption:

Here’s a suggested re-write of the Figure caption:

Fig. 1: Study site and seismicity. **a**) Distribution of seismic stations operating since January 2014 (black triangles) and May 2018 (green triangles). Red box marks the specific study region where we developed the earthquake catalog. **b**) Seismicity map from January 2014 to January 2023 showing strike-slip focal mechanisms (red/white circles) available from the National Earthquake Data Center plus earthquakes that occurred before (grey filled circles) and after (red filled circles) when the second landslide-dammed lake (LDL) almost reached peak water level. Red box marks the region shown in (d). **c**) Magnitude-frequency distribution of earthquakes from May 2018 to January 2023 within 10 km of the LDLs. Magnitude of completeness (M_c) and a - and b -values were calculated using the ZMAP software. **d**) Distribution of earthquakes from May 2018 to January 2023 within 10 km of the LDLs.

I would recommend re-writing all captions for clarity, as illustrated above.

Thanks for your suggestion. The orange line is LDLs - this is denoted in the figure legend. According to your suggestion, we have rewritten the figure captions:

Fig. 1 | Study site and seismicity. a Distribution of seismic stations operating since January 2014 (black triangles) and May 2018 (cyan triangles). Red box marks the specific study region where we developed the earthquake catalog. **b** Seismicity map from January 2014 to January 2023 (grey filled circles) showing strike-slip focal mechanisms (red/white circles) available from the National Earthquake Data Center³⁹ plus earthquakes that occurred during the week (10 to 16 November 2018) when the second landslide-dammed lake (LDL) reached peak water level (red filled circles). Red box marks the region shown in (d). **c** Magnitude-frequency distribution of earthquakes from May 2018 to January 2023 within 10 km of the LDLs. Magnitude

of completeness (M_c) and a - and b -values were calculated using the ZMAP software⁴⁰. **d** Distribution of earthquakes from May 2018 to January 2023 (grey filled circles) within 10 km of the LDLs plus earthquakes that occurred during the week (10 to 16 November 2018) when the second LDL reached peak water level (red filled circles).

Fig. 2 | Earthquakes triggered by 2018 Baige landslide-dammed lake. a Spatial distribution of Coulomb stress change (ΔCFS) at depth of 4.5 km corresponding to -0.5 km asl on 13 November 2018 and earthquakes from 10 to 16 November 2018 (green filled circles) within 10 km of the landslide-dammed lakes (LDLs). ΔCFS is from the combined effect of direct gravitational loading and pore pressure diffusion (Fig. 2d). **b** Cumulative number of earthquakes with local magnitudes greater than or equal to the magnitude of completeness ($M_c = 1.1$) and weekly seismicity rate over a five-year period with timings of the two 2018 Baige landslides (black arrows). Gray bar marks the period shown in (c and d). **c** Earthquake magnitudes and measured LDL water level^{35,36,37} over a fifty-day period. Gray dashed line marks M_c . **d** Temporal evolution of ΔCFS at point P (Fig. 2a).

Response to Reviewer #2:

Dear authors,

I wish to thank you for the hard and thorough work you have made to answer my comments as well as the comments of the other reviewers. The quality and robustness of the manuscript have both been significantly improved thanks to that. If the manuscript still relies on some debatable assumptions (as in most Earth Sciences paper), they are now clearly stated and well discussed or justified. Demonstrating the link between an LDL event and few potentially triggered earthquakes (of small magnitudes) is statistically difficult. Yet, the authors have made a real effort to strengthen this link by improving the quality of the catalog and by exploring in more details the parametrization of the processes (pore pressure diffusion and direct gravitational loading) which could trigger earthquakes in relation to LDL water level rise.

I therefore suggest the editor to accept the paper for publication after some minor revisions (see below).

Best regards,

Philippe Steer

We would like to thank you for the specific and constructive comments you provided for this manuscript. These comments are really helpful and greatly improve this manuscript.

Minor comments

- Abstract: please explicitly state (line 21) that the observed increase in seismicity rate after the LDL only applies to low magnitude earthquakes

Done.

...there was a statistically significant increase in earthquake activity (local magnitude ≤ 2.6)...

- Concerning pore pressure and its temporal evolution: Thank you for having added Fig. S16. However, I disagree with the interpretation that “the temporal evolutions of ΔCFS at a given point are similar for different hydraulic diffusivity, and that the peak seismicity rate coincides with the peak amplitude of ΔCFS assuming different hydraulic diffusivity (Fig. S16).” (as mentioned in your rebuttal letter and paper). Theory of pore pressure diffusion demonstrates that the pore pressure front progressively propagates from a point source at a distance d that is function of time t and Diffusivity D as $d = \sqrt{4 \pi D t}$ [see for instance Shapiro et al., 1997]. Therefore, diffusivity strongly impacts the timing of the arrival of the peak of pore pressure, as well as its amplitude, beneath the LDL. Fig. S16 (b and c) is also clearly showing that. You could strengthen your manuscript by 1) mentioning the range of D are consistent with having a DCFS above >0.01 MPa, and 2) by mentioning the relative contribution to DCFS of gravitational loading and pore pressure for these different values of D .

Thank you for your suggestion. According to your suggestion, we calculated the relative contribution to DCFS of pore pressure for these different values of D and added some description:

While hydraulic diffusivity strongly influences the time at which ΔCFS exceeds a certain threshold, we further confirm that ΔCFS exceeds 0.01 MPa for hydraulic diffusivities ranging from 0.02 to 2 m²/s. On the day of peak seismicity rate, the relative contribution to ΔCFS of pore pressure increases from ~0.1 to 0.9 as the assumed hydraulic diffusivity increases from 0.02 to 2 m²/s (Fig. S14).

Fig. S14 | Coulomb stress change (ΔCFS) and earthquakes triggered by 2018 Baige landslide-dammed lake. a Earthquake magnitudes and LDL water level over a fifty-day period. Gray dashed line marks M_c . **b** Temporal evolution of ΔCFS due to the combined effect of direct gravitational loading and pore pressure diffusion at point P (Fig. 2a) assuming different hydraulic diffusivity coefficient values. **c** Temporal evolution of ΔCFS due to pore pressure diffusion at point P (Fig. 2a) assuming different hydraulic diffusivity coefficient values. **d** Relative contribution to ΔCFS of pore pressure at point P (Fig. 2a) assuming different hydraulic diffusivity coefficient values.

• Declustering: the manuscript only considers the lower ranges of accepted parameters for declustering using the Reasenber approach (see for instance Table 3 of van Stiphout et al., 2012 <http://dx.doi.org/10.5078/corssa-52382934>). Please make additional tests with standard parameters and with the one of the lower range.

Done.

We also tested the impact of different input parameters on the declustering based on the approach from Reasenber⁴¹. We identified 4 clusters of earthquakes containing a total of 16 events (4 main earthquakes and 12 foreshocks/aftershocks) out of 62 events which occurred over ~5 years, using other suggested input parameters⁴¹: look-ahead time from 1 to 10 days, confidence probability of 0.95, effective min magnitude cutoff of 1.5, the increase in lower cutoff magnitude during clusters of 0.5, and the number of crack radii surrounding each earthquake of 10. While the declustered earthquake catalogs with different input parameters are different, we further confirm that this seismicity rate increase is statistically significant at a > ~90% level based on the above four statistical tests (Table S1).

Response to Reviewer #3:

The revised manuscript represents a significant improvement over the previous version, particularly in terms of the earthquake catalog and statistical analysis. However, I still have two principal concerns regarding the triggering mechanisms and earthquake statistics. While the manuscript has made notable improvements, addressing these concerns will enhance the clarity and credibility of the research findings.

We would like to thank you for the specific and constructive comments you provided for this manuscript. These comments are really helpful and greatly improve this manuscript. We have tried our best to resolve the issues you are still concerned about. We hope our revisions have addressed your concerns.

Man comments:

- (1) While the manuscript provides a more comprehensive earthquake catalog and statistics, the correlation between the spatiotemporal occurrence of earthquakes and pore pressure changes remains unclear. It is noteworthy that most events ($ML > 1.1$) occurred near the peak of the water level. However, the manuscript does not sufficiently elucidate the role of pore pressure throughout the entire process. Further clarification on how pore pressure influences seismic activity is warranted to enhance the understanding of the triggering mechanisms.

Thank you for your suggestion. We have added some description of the correlation between the spatiotemporal occurrence of earthquakes and pore pressure changes, and discuss the relative contributions from direct gravitational loading of LDL and pore pressure diffusion in triggering seismicity.

While gravitational loading can cause Coulomb stress to decrease in some areas, pore pressure always causes an increase in ΔCFS with its amplitude gradually decreasing away from the LDLs (Fig. S9). For earthquakes that occurred in regions where gravitational loading resulted in a decrease in Coulomb stress, it is clear that pore pressure is the primary mechanism triggering these events, though we neglect the coupling effect between pore pressure diffusion and gravitational loading^{24,25,26,27}. However, for those earthquakes that occurred in regions where gravitational loading resulted in an increase in Coulomb stresses, both gravitational loading and pore pressure may jointly control the triggered seismicity. After the second LDL formed, while the water level rises and seismicity begins to occur from 10 to 13 November (Fig. S2), the relative contribution to ΔCFS of gravitational loading decreases gradually from 0.45 to 0.41 (Fig. S13). Subsequently, after the dam breached on 13 November which is the day of peak seismicity rate, the water level rapidly decreased and the relative contribution to ΔCFS of gravitational loading decreases sharply from 0.41 to 0.19 so pore pressure becomes the dominant triggering mechanism. However, the relative contribution to ΔCFS of pore pressure depends on the assumed hydraulic diffusivity (Fig. S14). In addition, since the pore pressure diffusion is from a long winding river (LDLs) instead of a point source, there is no obvious migration pattern in seismicity with time (Fig. S15).

Fig. S13 | Relative contributions to Coulomb stress change (Δ CFS) of pore pressure and direct loading and earthquakes triggered by 2018 Baige landslide-dammed lake. a Earthquake magnitudes and LDL water level over a fifty-day period. Gray dashed line marks M_c . Relative contributions to Δ CFS of pore pressure (b) and direct loading (c) at point P (Fig.2a).

Fig. S15 | Coulomb stress change (Δ CFS) maps at depth of 4.5 km and earthquakes that occurred before (grey filled circles) and on (green filled circles) each day from 10 to 16 Nov 2018.

(2) It appears that the authors evaluate the statistical significance of the observed seismicity rate change by including events lower than the completeness magnitude (M_c). This approach may introduce biases and potentially obscure the true significance of seismicity rate changes. Revisiting the statistical analysis methodology to ensure that only events above the completeness magnitude are considered would strengthen the robustness of the findings and their interpretation.

Thanks for your comment. We apologize for the misunderstanding that may have occurred because we analyzed the seismicity rate without considering M_c , in addition to statistical tests on $\geq M_c$ and declustered events. To avoid confusion, we have now removed discussion regarding statistical analysis of the seismicity rate without considering M_c and only keep the discussion regarding statistical analysis on $\geq M_c$ and declustered events:

For each of $\geq M_c$ and declustered catalogs, we consistently find that the increase in seismicity rate is statistically significant at a $> 99\%$ level based on both the statistic P (ref.⁴²) and statistic Z (ref.⁴³) tests (Methods). We further confirm that the increase in seismicity rate is also statistically significant at a $>99\%$ level (Table S1) based on both the improved statistic based on Poisson probability⁴⁴ and an empirically derived statistic⁴⁵ (Fig. S6) tests (Methods) which were developed to determine statistically significant changes in earthquake rate under small background rate.

Table S1 | Seismicity rate change and statistical significance.

Time window	Seismicity rate per week		
	$\geq M_c$ events	Declassified events with input I^{**}	Declassified events with input II^{***}
1 week during the second LDL*	16	9	5
27 weeks before the second LDL (Total number of earthquakes)	0.3 (9)	0.3 (8)	0.3 (8)
224 weeks after the second LDL (Total number of earthquakes)	0.2 (37)	0.2 (37)	0.2 (37)
Maximum over observation period besides the second LDL	3	3	3
Statistical tests	Statistical significance		
	$\geq M_c$ events	Declassified events	Declassified events
Statistic P	100.00%	99.82%	97.41%
Statistic Z	100.00%	99.62%	96.52%
Improved statistic based on Poisson probability	100.00%	99.73%	89.92%
Empirically derived statistic	99.71%	99.71%	99.71%

* 1 week (10 to 16 November 2018) when the second LDL approached its peak water level

** look-ahead time from 0.5 to 3 days, confidence probability of 0.9, effective min magnitude cutoff of 1.1, the increase in lower cutoff magnitude during clusters of 0, and the number of crack radii surrounding each earthquake of 5

*** look-ahead time from 1 to 10 days, confidence probability of 0.95, effective min magnitude cutoff of 1.5, the increase in lower cutoff magnitude during clusters of 0.5, and the number of crack radii surrounding each earthquake of 10

Other comments:

1. Lines 121-124: By reviewing Figure 2 and the estimation of the number of events per week, it appears that the authors have included events with magnitudes lower than the completeness magnitude (M_c). As such, discussions regarding the seismic rate without considering M_c may lack meaningful interpretation. Therefore, I suggest removing these sentences for clarity and accuracy.

We have removed these sentences in the manuscript.

Figure 2:

Plot the cumulative number of earthquakes with magnitudes greater than or equal to M_c (1.1) in Figure 2b. This adjustment will provide a clearer representation of the seismic activity relevant to the study's objectives.

In Figure 2c, mark the value of M_c to highlight its significance in the context of the seismicity analysis. Additionally, plot the cumulative number of events since November 3 and compare the temporal evolution with the increase in water level. This comparison will offer valuable insights into the correlation between seismic activity and water level fluctuations.

We now plot the cumulative number of earthquakes with magnitudes greater than or equal to M_c (1.1) in Fig. 2b. We also marked M_c in Fig. 2c. Considering the layout of Figure 2, there is indeed no space to add a new panel. Therefore, we plot the cumulative number of events since November 3 and the temporal evolution of water level in Fig. S2a.

2. Lines 167-180: The manuscript discusses Coulomb failure stress changes in the context of triggered seismic events, which predominantly exhibit small magnitudes (ML less than 2). It is important to note that small events may not necessarily share similar geometries with adjacent fault systems.

Thank you for your suggestion.

Based on reviewer 1's comment, in the last round of revisions, we have checked that the polarities of two M2.6 and 1.9 are consistent with the focal mechanisms of those large events:

In addition, we find that the first motions recorded by surrounding seismic station of the two largest earthquakes (ML 2.6 and 1.9) as the second LDL approached peak water level are generally consistent with the left-lateral strike-slip focal mechanisms of the surrounding large earthquakes (Fig. S7).

Also, we agree with your concern that some small earthquakes may not necessarily share similar geometries with adjacent fault systems. Therefore, we added some discussion in the revised manuscript:

We assume a fault dip of 75° NE based on the focal mechanisms of four $M > 3.5$ earthquakes nearby and the consistency of the first motions of the two largest earthquakes (ML 2.6 and 1.9) as the second LDL approached peak water level with these focal mechanisms (Fig. S7), though we cannot confirm that the focal mechanism of the smaller earthquakes necessarily share similar geometries.

3. Line 179: I have observed that the hypocentral depths of many events in the dataset appear to be pinned at specified depths (Fig S8). Consequently, these events may not be suitable for estimating the median depth of triggered earthquakes.

Thanks for your comment. For events detected using template matching, their depths are assumed to be the same as the depths of the template. However, we have confirmed

that after removing events detected by template matching, the estimated median depth is still ~ 1.2 km (Fig. S8):

Similarly, for the earthquake catalog before template matching, the median depth of these triggered earthquakes is also ~ 1.2 km (Fig. S8).

Fig. S8 | Earthquake depths in complete catalog (a) and in catalog before template matching (b) over an eight-day period. Median earthquake depths in complete catalog and in catalog before template matching are 1.19 and 1.24 km, respectively.

- Lines 181-184: It appears that the authors solely estimated the increase in water level following the LSL, without accounting for the weight of the landslide sediments ($8.7 \times 10^6 \text{ m}^3$ as mentioned in Lines 98-99). Considering the substantial mass of the landslide sediments, which could exert gravitational loading, it is crucial to evaluate its potential contribution to induced seismicity alongside the rise in water level.

Thanks for your comment. Regarding the potential of landslide gravitational loading to trigger earthquakes, as reviewer 2 mentioned in the previous round of reviews, landslides do not represent a new source of mass, but only the very local redistributions of sediments along the landslide path. In our study case, these landslide sediments only traveled $\sim 1\text{-}2$ km that is even smaller than the mean horizontal location uncertainty (4 km) of our cataloged earthquakes, so we do not expect to be able to resolve the contribution of the redistribution of the landslide sediments in triggering seismicity. Furthermore, we admit that calculating the gravitational loading of landslide sediments is a challenge due to the lack of data regarding their spatiotemporal distribution. We also added some description:

Note that we ignore the ΔCFS resulting from gravitational loading of landslide sediments, since these landslide sediments are not new sources of mass but only the very local redistributions ($\sim 1\text{-}2$ km) of sediments along the landslide sliding path.

- Lines 227-229: How to estimate the temporal evolution of water level rise after the formation of LDL?

The temporal evolution of water level rise at the dam is from field measurements. After these two landslides, the Chinese government immediately launched emergency observation and response. There are some studies on these two LDLs that present these

field measurements. We have added these references in the revised manuscript.

Zhong, Q., Chen, S., Wang, L. & Shan, Y. Back analysis of breaching process of Baige landslide dam. *Landslides* **17**, 1681-1692 (2020).

Gao, Y., Zhao, S., Deng, J., Yu, Z. & Rahman, M. Flood assessment and early warning of the reoccurrence of river blockage at the Baige landslide. *Journal of Geographical Sciences* **31**, 1694-1712 (2021).

6. Lines 182-183: The manuscript lacks clarity regarding how the authors evaluate the contributions from direct gravitational loading of LDL and fluid pressure diffusion to induced seismicity. Despite the reported rise in water level, there is no discernible spatiotemporal migration of earthquakes associated with this increase. Notably, most events cluster near the peak of the water level, raising questions about the precise mechanisms driving seismic activity. Given the absence of clear migration patterns and the limited magnitudes of events, it remains uncertain whether the observed seismicity results from direct gravitational loading, fluid pressure diffusion, or other factors.

Thanks for your comment. Similar to your comment above, to address your concerns, we have added the following description and discussion:

While gravitational loading can cause Coulomb stress to decrease in some areas, pore pressure always causes an increase in ΔCFS with its amplitude gradually decreasing away from the LDLs (Fig. S9). For earthquakes that occurred in regions where gravitational loading resulted in a decrease in Coulomb stress, it is clear that pore pressure is the primary mechanism triggering these events, though we neglect the coupling effect between pore pressure diffusion and gravitational loading^{24,25,26,27}. However, for those earthquakes that occurred in regions where gravitational loading resulted in an increase in Coulomb stresses, both gravitational loading and pore pressure may jointly control the triggered seismicity. After the second LDL formed, while the water level rises and seismicity begins to occur from 10 to 13 November (Fig. S2), the relative contribution to ΔCFS of gravitational loading decreases gradually from 0.45 to 0.41 (Fig. S13). Subsequently, after the dam breached on 13 November which is the day of peak seismicity rate, the water level rapidly decreased and the relative contribution to ΔCFS of gravitational loading decreases sharply from 0.41 to 0.19 so pore pressure becomes the dominant triggering mechanism. However, the relative contribution to ΔCFS of pore pressure depends on the assumed hydraulic diffusivity (Fig. S14). In addition, since the pore pressure diffusion is from a long winding river (LDLs) instead of a point source, there is no obvious migration pattern in seismicity with time (Fig. S15).

Figure 1 Cayan or Green triangles? Gray and red dots are too big.

We've changed "blue" to "cyan" and made the gray and red dots smaller.

Figure 2 See my earlier comment

Done.

REVIEWERS' COMMENTS

Reviewer #3 (Remarks to the Author):

The authors have addressed all of my comments. The paper can be accepted in its present form.

Response to Reviewer #3:

The authors have addressed all of my comments. The paper can be accepted in its present form.

We would like to thank you for the specific and constructive comments you provided for this manuscript. These comments are really helpful and greatly improve this manuscript.